# Social-like responses are inducible in asocial Mexican cavefish despite the exhibition of strong repetitive behavior

Motoko Iwashita, Masato Yoshizawa*

School of Life Sciences, the University of Hawai'i at Manoa, Honolulu, United States

**Abstract** Social behavior is a hallmark of complex animal systems; however, some species appear to have secondarily lost this social ability. In these non-social species, whether social abilities are permanently lost or suppressed is unclear. The blind cavefish *Astyanax mexicanus* is known to be asocial. Here, we reveal that cavefish exhibited social-like interactions in familiar environments but suppressed these interactions in stress-associated unfamiliar environments. Furthermore, the level of suppression in sociality was positively correlated with that of stereotypic repetitive behavior, as seen in mammals. Treatment with a human antipsychotic drug targeting the dopaminergic system induced social-like interactions in cavefish, even in unfamiliar environments, while reducing repetitive behavior. Overall, these results suggest that the antagonistic association between repetitive and social-like behaviors is deeply shared from teleosts through mammals.

## Introduction

Many animals across a wide range of taxa express collective behaviors (e.g. shoaling and herding) resulting in multiple benefits, including foraging and/or antipredator effects (*Pitcher and Parrish, 1993*). Despite the benefits of collective behavior, this type of behavior in some species has been reduced over evolutionary time such that these species exhibit marked asociality when compared to their socially closest relatives (*Pitcher and Parrish, 1993*). Additionally, even among social species, this collective behavior can disappear when the animal is exposed to a stress-associated environment (*Kim et al., 2016*; *Lewis et al., 2007*). Instead, these animals may express asocial behavior and/or stereotypic repetitive behaviors, which have been particularly well characterized in mammals (e.g. 'zoochosis') (*Lewis et al., 2007*; *Rose et al., 2017*). Stereotypic repetitive behaviors, in contrast to collective behaviours, are generally thought of as non-functional (*Lewis et al., 2007*). Psychiatric conditions in humans, such as autism spectrum disorder (ASD), share similarities to atypical repetitive behavioral responses in other animals in that social activity is decreased and there is an overall increase in stereotypic repetitive behaviors, especially in stress-inducing environments (*Kaplan and McCracken, 2012*; *Langen et al., 2011a*; *Runco et al., 1986*). The neural mechanisms underlying the contrasting expression of social and repetitive behaviors and how the 'choice' between stereotypic repetitive and social modes of behavior are made remain largely unknown (*Kazdoba et al., 2016*; *Kim et al., 2016*; *Langen et al., 2011b*; *Lewis et al., 2007*). However, studies in mammals have hinted at such neural mechanisms by indicating that the dopaminergic striatal system is key for behavioral choices. For example, treatment with anti-agitation drugs (i.e. drugs to reduce repetitive behavior, for example aripiprazole and risperidone) attenuated but did not eliminate the activity of the D2-dopamine receptor-based indirect pathway in mammals (*Grillner, 2018*; *Kim et al., 2016*). Importantly, this core dopaminergic system exists in all vertebrate models from lampreys, teleosts (zebrafish), amphibians (*Xenopus*), and birds (chickens) to mammals (mice and humans) (*Grillner and Robertson, 2016*; *Stephenson-Jones et al., 2011*). However, among actinopterygians (ray-fin vertebrates), it is largely unknown whether an evolved asocial population had lost the ability to congregate with each

**\*For correspondence:**
yoshizaw@hawaii.edu

**Competing interest:** The authors declare that no competing interests exist.

other or just suppressed this activity and whether repetitive behaviors are enhanced under stress in an antagonistic manner with collective behavior.

To address these two questions, we used a suitable vertebrate model, the Mexican tetra *Astyanax mexicanus*, and developed a novel experimental workflow to assay collective and stereotyped behavior (*Yoshizawa, 2015*). *A. mexicanus* is emerging as a key model system for the study of diverse aspects of evolution and development, including those with relevance to human medicine, for example cataract formation, obesity, diabetes, albinism-related syndrome, and insomnia (*Aspiras et al., 2015*; *Bilandžija et al., 2018*; *Bilandžija et al., 2013*; *Duboué et al., 2012*; *Duboué et al., 2011*; *Jaggard et al., 2017*; *Keene et al., 2016*; *Ma et al., 2014*; *McGaugh et al., 2014*; *Riddle et al., 2018*; *Rohner et al., 2013*; *Strickler et al., 2007*). *A. mexicanus* comprises surface riverine epigean (surface fish) and cave-dwelling hypogean (cavefish) forms. Cavefish diverged from their surface-dwelling relatives 20,000~200,000 years ago (*Fumey et al., 2018*; *Herman et al., 2018*) and since then have evolved many distinct morphological and behavioral phenotypes, including eye regression/loss, pigment reduction, increased number of lateral line mechanosensors, enhanced behavior responses to vibrational stimuli, sleeplessness, hyperactivity and repetitive circling (*Keene et al., 2016*; *Yoshizawa, 2015*; *Yoshizawa et al., 2018*). Compared to cavefish, surface fish show typical fish phenotypes, including eyed and pigmented morphologies, no strong sensitivity to vibrational stimuli, and nocturnal sleep patterns. Cavefish have also been shown to have reduced social-like interactions, such as no detectable schooling and shoaling behavior (*Kowalko et al., 2013*; *Patch et al., 2020*; *Pierre et al., 2020*) or hierarchal dominance (*Elipot et al., 2013*). In contrast, surface fish school/shoal with cohorts and model fish (*Kowalko et al., 2013*) show group hierarchical dominance (*Elipot et al., 2013*). Importantly, the surface fish schooling behavior was concluded to be driven by visual stimuli because it diminished in the dark (*Kowalko et al., 2013*). In contrast, surface fish in the dark continued to show similar levels of aggression and were thought to establish a hierarchical dominance, similar to those kept in the light (*Elipot et al., 2013*). This motivated us to test other social-like interactions in the dark to compare non-visual-based social-like interactions between surface fish and cavefish.

Thus, we developed a sensitive method to detect social-like interactions and repetitive behavior in the dark (*Figure 1*). This method revealed a number of social-like interactions in surface fish in the dark. Surface fish, however, did not show repetitive turning. Furthermore, we also revealed weak but existent social-like interactions in cavefish that occurred while exhibiting a significantly higher level of repetitive behavior than that of surface fish. During these nearby interactions, cavefish decelerated their swimming speed and were likely to follow their cohorts. The comparison of these behaviors under familiar and potentially stressful, unfamiliar environments indicated that cavefish showed significantly more nearby interactions in the familiar environment. The atypical dopamine psychiatric drug aripiprazole, which is known to increase social behaviours and reduce repetitive motions in humans (*Coleman et al., 2019*; *de Bartolomeis et al., 2015*; *Ghanizadeh et al., 2014*), reduced swimming speed and increased nearby interactions in cavefish. Cavefish also showed reduced levels of repetitive turning after aripiprazole treatment. In summary, cavefish showed robust repetitive circling, and the degree of circling was negatively correlated with social-like nearby interactions. This cavefish interaction remained plastic and was suppressed under a potentially stressful environment. Together with the reported parallels in pharmacological responses, a set of behavioral symptoms, and gene expression profiles between cavefish and humans (*Yoshizawa et al., 2018*), the presented results suggest that the asocial cave population is still capable of performing social-like nearby interactions under environmental or pharmacological treatments and shares an antagonistic association between social-like and repetitive behaviors with mammals.

## Results

### Detecting nearby interactions in a non-visual, infrared-illuminated condition

Since there are few reports on social-like interactions of teleost species in the dark (e.g. *Gerlai, 2014*), we began by systematically assessing close-distance interactions using social and non-social morphs of *Astyanax mexicanus*—surface fish and cavefish—in a non-visual setup. Fish were introduced into an unfamiliar environment, the recording arena, where we could observe nearby interactions at close range (see below). The behaviors of four fish were recorded for 5 min (*Figure 1*; *Videos 1 and 2*) in

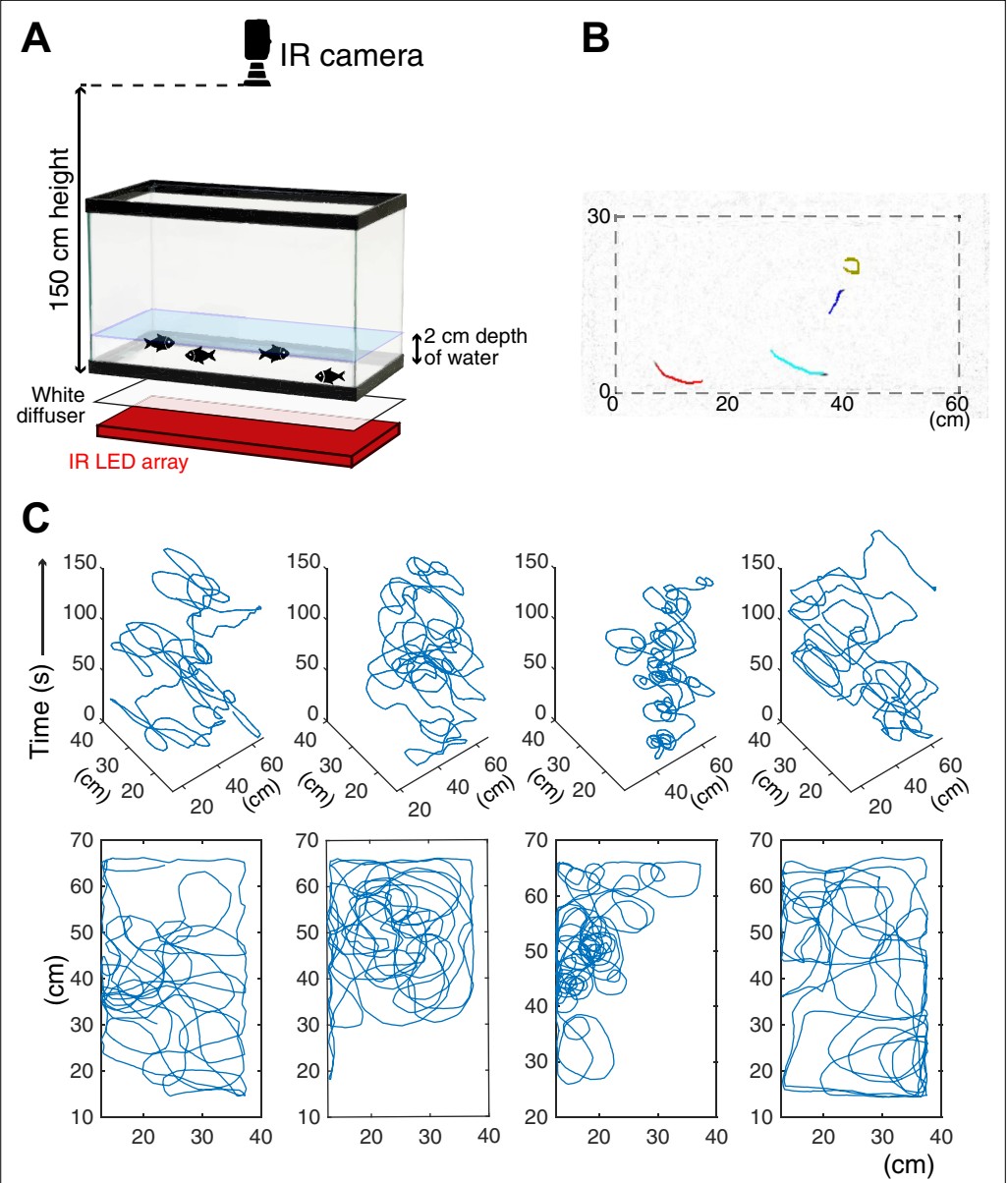

**Figure 1.** Schematics of the behavioral assay. (**A**) Schematics of the assay system. The recording tank was back-lighted by an infrared (IR) LED array with a white paper diffuser. The IR camera was set at the top of the tank, and fish movement was recorded. (**B**) One example frame of recorded cavefish video. Coloured lines represent the 1 s (20 frames) trajectories of individual fish detected by idTracker software. (**C**) Example trajectories of each cavefish from a 2.5 min recording. Each fish shape's ID was determined by idTracker, and its X-Y coordinates were traced across the entire recording time. The top row (four panels) shows the X-Y-time traces of each of four fish. The bottom row represents the X-Y traces by stacking the time series of the corresponding top row panel.

The online version of this article includes the following figure supplement(s) for figure 1:

**Figure supplement 1.** No visual response to the back light emitted from the 850 nm infrared (IR) LED.

**Figure supplement 1—source data 1.** Number of fish found in the infrared-lighted or shaded area within a 5 min period.

a black box that was evenly back-lighted with an 850 nm light-emitting diode (LED) (Materials and methods). The individual trajectory of each of the four individual fish was tracked by idTracker (**Pérez-Escudero et al., 2014**). To verify whether fish visually responded under 850 nm lighting, surface fish and cavefish were tested for phototaxis and startle responses when presented with a moving

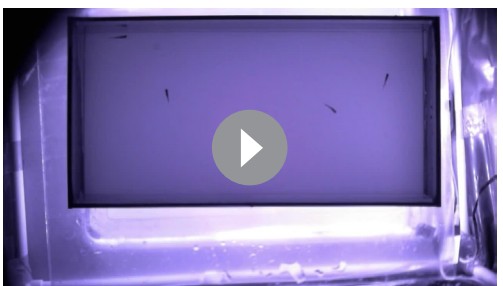

**Video 1.** Swimming patterns of four surface fish in dark conditions (unfamiliar environment).
https://elifesciences.org/articles/72463/figures#video1

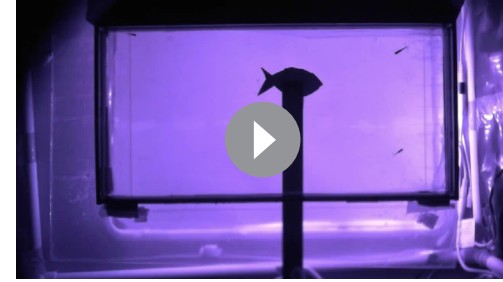

**Video 3.** Swimming patterns of four surface fish in dark conditions. The arena was illuminated with infrared light. Fish did not respond to the visual stimulus—a moving object (unfamiliar environment).
https://elifesciences.org/articles/72463/figures#video3

shade (*Elipot et al., 2013*; *Kowalko et al., 2013*). Neither surface fish nor cavefish showed photo-taxis or a preference for illuminated or non-illuminated areas (*Figure 1—figure supplement 1*). In addition, sighted surface fish did not avoid the shade approaching them (*Video 3*). These results suggested that neither surface fish nor cavefish visually responded under 850 nm illumination. We call this non-visual setup the 'dark' condition hereafter.

Surface fish are known to school/shoal in lighted environments (*Kowalko et al., 2013*; *Wilkens et al., 1988*) and exhibited strong shoaling and schooling in our setup in lighted conditions (*Video 4*; see below). In the dark, two standard shoaling measurements (*Miller et al., 2013*; *Partridge et al., 1980*; *Pitcher, 1973*)—the averages of nearest neighbor distance (NND: the distance between a focal individual and its nearest neighbor) and inter-individual distance (IID: the mean distance between each individual)—of surface fish were indistinguishable from those of cavefish (*Figure 2A and B*). However, we observed differences in how cohorts of cavefish and surface fish interacted; in particular, surface fish changed their swimming directions and followed each other (*Figure 2C*), while cavefish showed this pattern less frequently. Due to the relatively low shoaling tendency of surface fish in the dark and the difficulty of detecting interactions between any two arbitrary fish in IID and NND, a new measurement of this social-like interaction was needed. Here, we define 'nearby interaction' as a few fish coming within a short distance of each other and then staying close for some period at a level higher than that expected by chance. These criteria can suggest intent in the movement rather than random locomotion with respect to their conspecifics. First, to determine the criteria for the maximum distance (cut-off distance) and the minimum duration (cut-off duration) to use in defining a bout of nearby interaction, we employed random sampling by choosing four fish among 18 groups (each group consisted of 4 surface fish). These simulated random data enabled us to validate whether fish were closer to each other than by random expectation. A total of 1280 random samples were performed with the surface fish (see Materials and methods). By using the X-Y coordinates of randomly sampled surface fish, the IID in each frame was calculated and plotted as a histogram, and this was considered the simulated no-interaction conditions (the blue lines in *Figure 2D*). The IID instead of NND was strategically chosen because we were

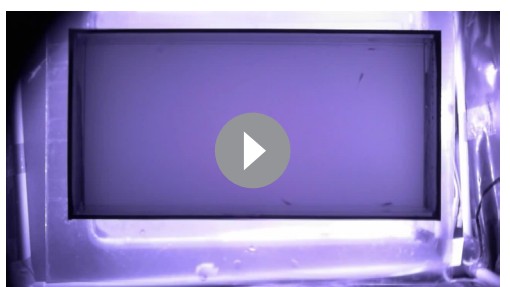

**Video 2.** Swimming patterns of four cavefish in dark conditions (unfamiliar environment).
https://elifesciences.org/articles/72463/figures#video2

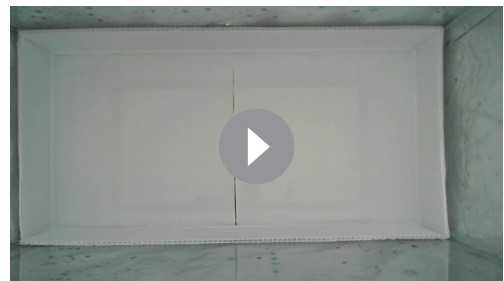

**Video 4.** Swimming patterns of four surface fish in lighted conditions (unfamiliar environment).
https://elifesciences.org/articles/72463/figures#video4

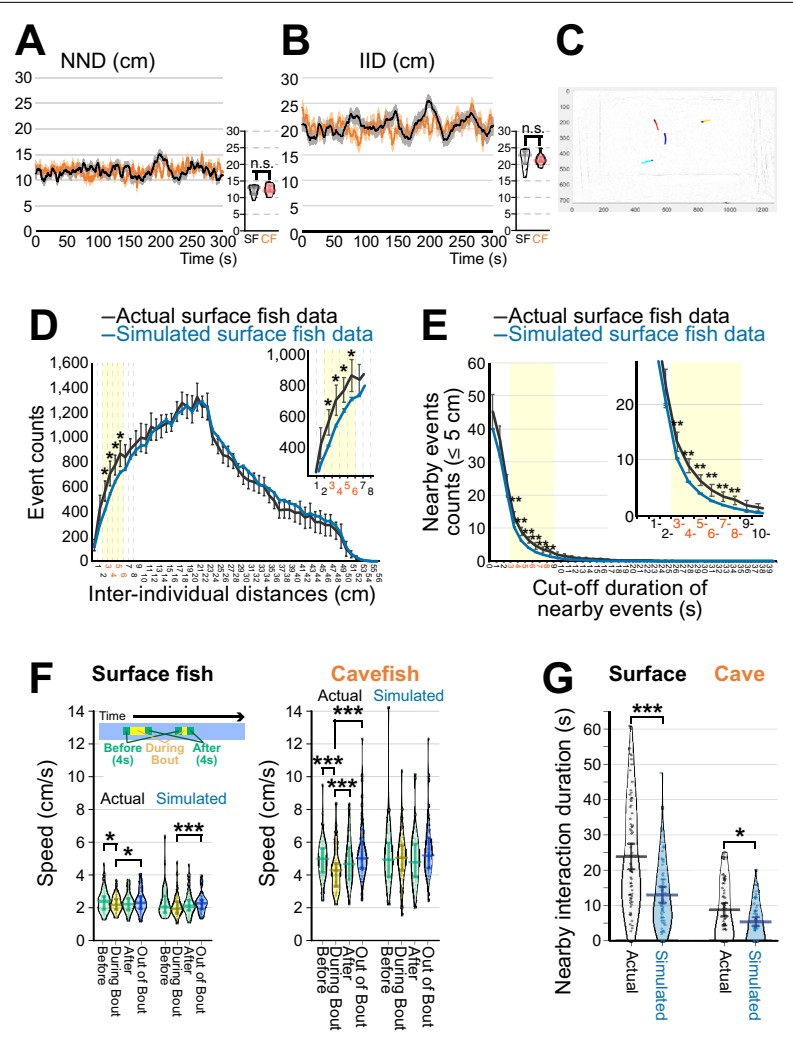

**Figure 2.** Detection of social-like nearby interactions. (**A, B**) Measurements of nearest neighbour distance (NND) (**A**) and interindividual distance (IID) (**B**). Surface fish are shown as black lines, and cavefish are shown as orange lines (N = 18 groups of four each). The means ± standard errors of the mean (s.e.m.) in each frame are shown. Smaller panels on the right side of each of (**A**) and (**B**) show averages of NND or IID sampled every 30 s of the 300 s assays. n.s.: not significant. (**C**) An example frame of recorded surface fish video. Coloured lines represent trajectories of individual fish detected by idTracker software (1 s/20-frame trajectories). A red-labeled fish was followed by a blue-labelled fish. (**D**) Distribution of IID across all frames of 18 surface fish groups, where each group yields six IIDs from each pair among the four fish ($_4C_2 = 6$, black line: means ± 95 % confidence intervals (CIs)). Similarly, the distribution of IIDs was calculated from 1,280 simulated randomly sampled groups (blue line: means ± 95% CI). IID between 3 and 6 cm showed significant separation between actual and simulated IIDs (yellow shaded areas). Inset shows the magnified 3–6 cm interval. *: p < 0.05, t-test between the actual and simulated data adjusted by the Holm correction within the yellow shaded area. (**E**) Inverse cumulative distribution of the nearby-event durations. The inverse cumulative distributions for the event durations whose IIDs were less than 5 cm are shown, where the black lines represent the results from the actual 18 surface fish groups (means ± 95% CI) and the blue lines represent the results of the 1,280 simulated groups (means ± 95% CI). In the actual surface fish data, the nearby-interaction durations between 3 and 8 s were significantly higher than the simulated surface fish data. **: p < 0.01, t-test between the actual and simulated data adjusted by the Holm correction within the yellow shaded area. (**F**) Swimming velocity during or out of nearby interactions. The mean swimming speeds (1) during the 4 s before the nearby-interaction bout, (2) during the bout, (3) during 4 s after the bout, and (4) during out-of-bout periods are shown for surface fish (left) and cavefish (right) by using the actual and simulated dataset (N = 72 from 18 groups for each of surface fish and cavefish and for each of the actual and randomly sampled data). Swimming speeds are lower in the cavefish during the bout in the actual data. (**G**) Pirate plots depicting nearby-interaction durations detected by the newly developed method. The plot of the actual data is shown in black and white, and

*Figure 2 continued on next page*

*Figure 2 continued*

the simulated data are shown in light blue (N = 72 from 18 groups for each of the surface fish and cavefish). The thicknesses of the beans represent data density. *: p < 0.05, ***: p < 0.001. All statistical scores are available in *Supplementary file 1*. The new detection protocol for determining nearby interactions is described in Materials and methods and *Figure 2—figure supplements 1–3*. The involvement of the lateral line sensory system in the nearby interaction was tested, and the data are presented in *Figure 2—figure supplement 4*.

The online version of this article includes the following source data and figure supplement(s) for figure 2:

**Figure supplement 1.** Determine the cut-off distance and cut-off duration for detecting nearby interactions.

**Figure supplement 2.** Details of nearby interactions were obtained by classifying them into five categories.

**Figure supplement 3.** Nearby interactions of surface fish in lighted conditions.

**Figure supplement 3—source data 1.** The X-Y coordinates (cm) and distances of each pair (cm) of a four-surface fish group under the light.

**Figure supplement 4.** Pharmacological ablation of the lateral line by gentamicin antibiotics reduced nearby interactions.

**Figure supplement 4—source data 1.** Nearby interaction duration (s) under the gentamicin treatment.

**Source data 1.** Surface fish's and cavefish's inter-individual distance (cm) in every 0.05 s.

**Source data 2.** Surface fish's and cavefish's inter-individual distance (cm) in every 0.05 s.

**Source data 3.** Event numbers counted under the cut-off distances between pairs of fish in the actual and simulated surface fish data.

**Source data 4.** Nearby event numbers shorter than 5 cm distance counted more than cut-off duration between pairs of fish in the actual and simulated surface fish data.

**Source data 5.** Speed (cm/s) during the nearby interaction events and other periods, and the detected nearby interaction duration (s) comparing between the actual and simulated random data.

**Figure supplement 1—source data 1.** The X-Y coordinates (cm) and distances of each pair (cm) of a four-surface fish group in the dark.

**Figure supplement 1—source data 2.** The X-Y coordinates (cm) and distances of each pair (cm) of a four-cavefish group in the dark.

---

interested in interactions not only between two fish (NND) but also among three or four fish that were closer than expected by chance. The actual surface fish IID scores ranged from 0.0 to 55.0 cm and showed significant differences in event counts between the 3.0- and 6.0 cm cut-offs compared to the simulated surface fish IID scores (*Figure 2D*). We then selected the second longest distance, 5.0 cm, as a relatively conservative cut-off distance (*Figure 2D*; *Figure 2—figure supplement 1A*). Using this cut-off distance, we surveyed the minimum cut-off duration, ranging from 0.0 to 40.0 s (*Figure 2E*). The minimum cut-off durations ranging from 3.0 to 8.0 s with the actual surface fish data resulted in significantly more nearby-interaction event counts than those of the simulated surface fish data. We thus chose the second shortest duration, 4 s, as a relatively conservative minimum cut-off duration (*Figure 2E*; *Figure 2—figure supplement 1B*). With these criteria ( ≤ 5 cm and ≥4 s) defining nearby interactions, we standardized nearby interactions by subtracting passing-by durations (PbDur) calculated from the average swimming speed of each group (*Figure 2—figure supplement 1C*; Materials and methods). With these criteria and methods, nearby interactions were detected in both surface fish and cavefish (*Figure 2—figure supplement 1B,D, E, G* ).

Next, we evaluated whether these detected interactions were reciprocal rather than unidirectional, as seen in aggression and/or foraging. First, if the detected nearby interactions included aggression/foraging activities, the fish would accelerate towards the target or accelerate to escape (personal observation)(*Elipot et al., 2013*). To test this possibility of acceleration, shifts in speed were assessed by comparing speeds (i) during a 4 s period directly before the bout of a nearby interaction, (ii) during the bout, (iii) during a 4 s period after the bout, and (iv) during the rest of the recording time. Remarkably, both surface fish and cavefish slowed rather than accelerated during the nearby-interaction bouts compared to before the bouts (*Figure 2F*; *Supplementary file 1*). This speed shift from before to during the bout was not detected in the simulated data (*Figure 2F*; *Supplementary file 1*). Note that no sign of fin or scale damage was observed after each of the nearby-interaction assays, which has previously been reported in *A. mexicanus* in situations involving foraging or aggression against

intruders (*Elipot et al., 2013*). Second, we visualized reciprocal interactions during the nearby interaction by categorizing the patterns of gathering and dispersing into five categories: two fish disperse, two fish swim by in an inverse direction, two fish gather, fish one leads, and fish two leads (*Figure 2—figure supplement 2A* and B). The raster plots show switches between the leader and follower within each bout in the same pairs (yellow and orange); in the raster plots, we colour-coded these 5 types of interaction categories (*Figure 2—figure supplement 2C* and E). These switches were observed in both surface fish and cavefish and were also observed in schooling surface fish in the lighted condition (*Figure 2—figure supplement 3*; *Video 1*, *Video 2* and *Video 4*). Taken together, we defined nearby interactions as reciprocal social-like interactions, which were observed in both surface fish and cavefish in the dark.

Regarding these nearby interactions (shorter than 5.0 cm distance and more than 4.0 s duration), both surface fish and cavefish showed significantly longer durations in these interactions than randomly sampled simulated data (*Figure 2G*). The actual during-bout speeds in both the surface fish and cavefish were lower than speeds out of the bouts (*Figure 2F*). Cavefish speed during the bout was also significantly lower than the after-bout speed (right panel: *Figure 2F*). In contrast, in the simulated data, the swimming speed during a bout was indistinguishable from other speeds in cavefish (left panel: *Figure 2F*) and both before- and after-bout speeds in surface fish (*Figure 2F*). The surface fish during-bout speed was, however, lower than the out-of-bout speed in the simulated data, which we suggest was due to arena constraints and is discussed later (*Figure 2F*). These results suggested that not only surface fish but also 'asocial' cavefish can express social-like interactions in the dark at a significantly higher than random chance level.

## Involvement of lateral line mechanosensors in nearby interactions in the dark

Previous research has strongly suggested that the lateral line is the major sensor controlling shoaling/schooling among non-visual sensory systems (*Bleckmann, 1986*; *Montgomery et al., 2013*). To tested the involvement of lateral line mechanosensors, we employed a method of pharmacological ablation of the lateral line by applying gentamicin aminoglycoside antibiotics—an established method for ablating the lateral line sensors of teleosts, including *A. mexicanus* (*Figure 2—figure supplement 4*; *Yoshizawa et al., 2010*). Gentamicin treatment diminished the functional neuromasts—the sensory units of the lateral line labelled by a DASPMI vital dye—in both surface fish and cavefish (*Figure 2—figure supplement 4B-E*; *Van Trump et al., 2010*; *Yoshizawa et al., 2010*) and reduced the duration of nearby interactions in surface fish (p = 0.0112; *Figure 2—figure supplement 4A*). In contrast, there was no detectable change in nearby interactions in cavefish (p > 0.05, *Figure 2—figure supplement 4A*), although the neuromasts were ablated in a manner similar to surface fish (*Figure 2—figure supplement 4B-E*). Accordingly, this result suggested that lateral line sensors contributed to social-like interactions in surface fish in the dark.

## Effects of familiar and unfamiliar environments on nearby interactions

In mammals, social tendencies are suppressed in novel unfamiliar environments (*Boulter et al., 2014*; *Buhr and Dugas, 2009*; *Kim et al., 2016*). We therefore tested whether a similar behavioral response could be observed in the two *Astyanax* morphs by measuring the level of social-like interactions in response to familiar and unfamiliar environments (*Figure 3A*).

In both the familiar and unfamiliar environments, after their introduction to the recording arena, surface fish dispersed and swam slightly more often in the periphery than in the centre of the arena, which may represent thigmotaxis (*Figure 3B and C*; the position indices were below 0.5) (*Patton et al., 2010*). Cavefish also dispersed following transfer (*Figure 3C*) but showed a greater swimming distance than surface fish (*Figure 3D*). Within 1 min after release, surface fish started intermittently following nearby cohorts. Cavefish followed their cohorts less frequently than surface fish (see below). Notably, surface fish had significantly lower swimming distances in the unfamiliar environment than in the familiar environment, but the swimming distance in cavefish was not altered based on the environments (*Figure 3D*). Implementing the novel metric for nearby interactions, analysis of social-like interactions revealed that surface fish and cavefish responded differently to unfamiliar environmental stimuli (*Figure 3F and G*; Pop× Env: p < 0.0004 and p < 0.0095, respectively; *Supplementary file 1*). In surface fish, the unfamiliar environment significantly increased nearby-interaction duration and

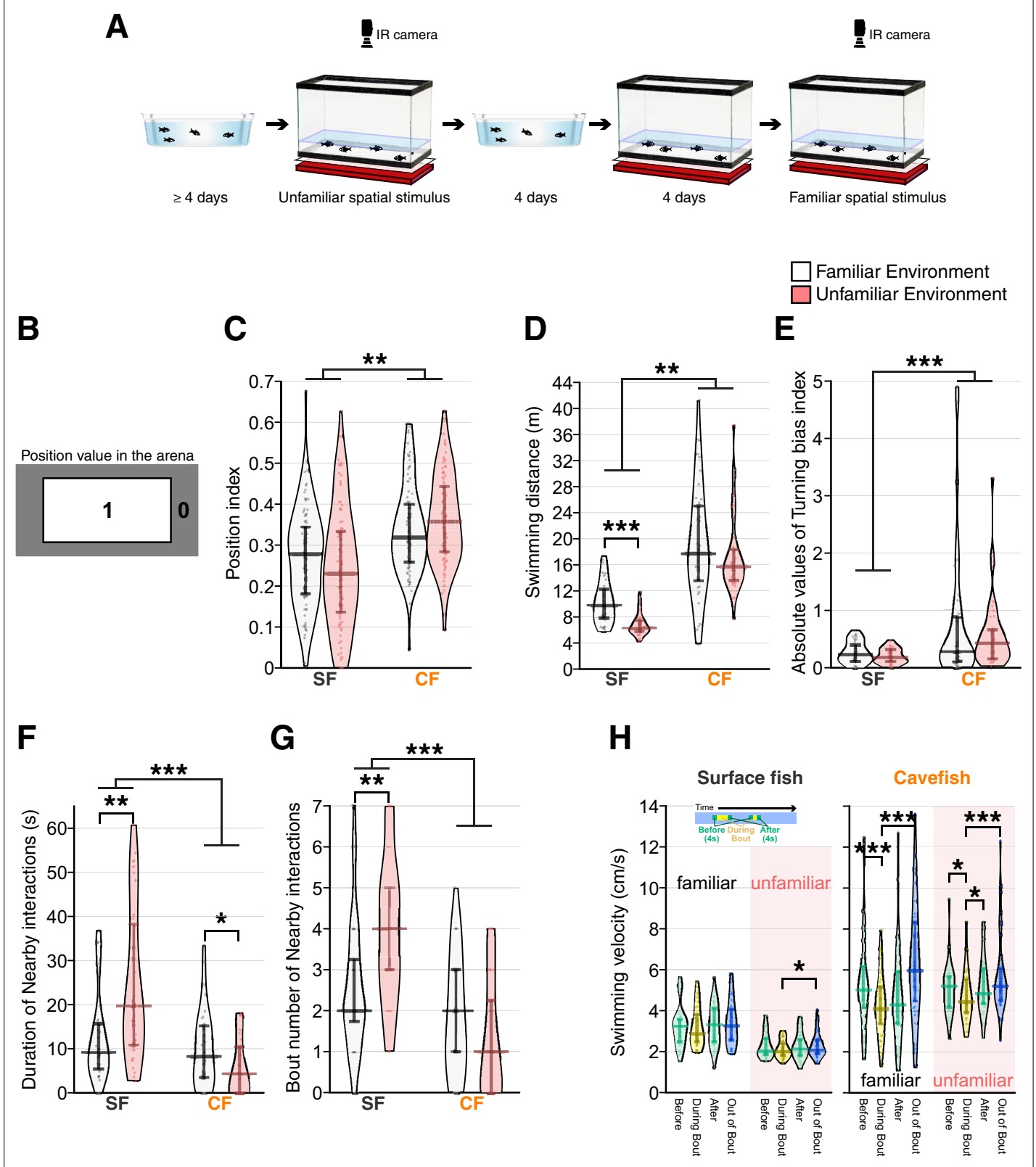

**Figure 3.** Reduced social-like nearby interactions in cavefish in unfamiliar environments compared with familiar environments. (**A**) Schematics of experimental procedure. Fish were acclimated in the home tray for more than 4 days and were transferred to the recording tank in this unfamiliar environment. After the recording, fish were returned to the home tray where they remained for more than 4 days. Then, the fish were returned to the recording tank and acclimated to the recording tank for 4 days. After acclimation, an infrared (IR) LED array was set to illuminate fish, and fish

*Figure 3 continued on next page*

*Figure 3 continued*

movements were recorded in this familiar environment. (**B**) The position index to characterize location preference (top view of the recording arena). Region one represents the centre, and region 0 represents the peripheral area of the recording. The areas of both region 1 and 0 are exactly the same. (**C**) Plots for the positional preference of the fish for region 1. The longer bars at the middle of the beans represent medians, and shorter bars represent 25th (bottom bar) and 75th (top) percentiles. Positional preference showed no significant change between the familiar and unfamiliar environments, but surface fish preferred the peripheral area more than cavefish. (**D**) Swimming distance plots indicated that cavefish swam significantly longer distances than surface fish. Surface fish had lower swimming distances in the unfamiliar environment. (**E**) The turning bias index indicated that cavefish turned more unidirectionally than surface fish, and the environmental shift did not affect the turning bias. (**F, G**) The duration (**F**) and the number of nearby-interaction bouts: both scores supported a significantly higher level of nearby interactions in surface fish. Notably, the nearby-interactions durations (**G**) showed similar scores in the familiar environment between surface fish and cavefish, yet the unfamiliar stimulus increased (surface fish) or decreased (cavefish) the duration, resulting in a significant Env× Pop interaction in the generalized linear model (*Supplementary file 1*). (**H**) Swimming velocity during or out-of-nearby interactions. The mean swimming speeds (1) during 4 s before the nearby-interaction bout, (2) during the bout, (3) during 4 s after the bout, and (4) during the out-of-bout period are plotted for surface fish (left) and cavefish (right) in the familiar and nonfamiliar environments. The average speed was reduced during the bout in cavefish under both familiar and unfamiliar environments. In contrast, the surface fish changed their speed profile only in the unfamiliar environment. SF: surface fish (N = 36 from nine groups); CF: cavefish (N = 36 from nine groups). **: p < 0.01, ***: p < 0.001. All detailed statistical scores are available in *Supplementary file 1*.

The online version of this article includes the following source data for figure 3:

**Source data 1.** Position-indices in the recording arena under the familiar or unfamiliar environment.

**Source data 2.** The swimming distances, absolute values of the turning indices, nearby interaction durations, nearby interaction event numbers, and swimming speeds under the familiar or unfamiliar environment.

bout number (*Figure 3F and G*; $r = 0.388$, $p = 0.0021$ and $r = 0.381$, $p = 0.0025$). In contrast, the unfamiliar environment reduced nearby-interaction durations in cavefish (*Figure 3F and G*; $r = 0.243$, $p = 0.0397$ and $r = 0.200$, $p = 0.09$; *Supplementary file 1*). Notably, in the familiar environment, the levels of nearby-interaction durations of surface fish and cavefish were comparable, which suggested that social-like nearby interaction is inducible in cavefish.

Regarding swimming speed during nearby interactions (*Figure 3H*), surface fish were more active in the familiar environment than in the unfamiliar environment, as shown in the swimming distance scores (*Figure 3D*). However, surface fish did not reduce their swimming speed during the bout compared to other time periods (the only exception was the out-of-bout speed in the unfamiliar environment; *Figure 3H*). Considering the difference from the previous surface fish experiment performed in the unfamiliar environment (actual data in *Figure 2F*), where surface fish significantly reduced the during-bout speed compared with the before-bout speed, surface fish did not consistently reduce their speed in the nearby interactions, yet speed reductions were occasionally detected.

The speed profile of the cavefish did not change between the familiar and unfamiliar environments (speed: $F_{3,198} = 5.7$, $p = 0.0009$; env (fam vs unfam): $F_{1,198} = 0.3$, $p = 0.5655$; speed× env: $F_{3,198} = 0.8$, $p = 0.4825$); however, the swimming velocity of cavefish significantly reduced during the interaction bout, suggesting that they responded to their cohorts, regardless of their familiarity with the environment (*Figure 3H*; *Supplementary file 1*). In summary, surface fish and cavefish differentially responded to environmental stimuli, and cavefish expressed a comparable level of nearby interactions to surface fish in a familiar environment.

## Turning bias index as a measure of repetitive circling and its relationship to nearby interactions

In mammals, social tendencies seem to be suppressed in those who show stereotypic repetitive motions (*Andari et al., 2010*; *Hollander et al., 2003*; *Hong et al., 2014*; *Kim et al., 2016*; *Langen et al., 2011b*; *Langen et al., 2011a*; *Lewis et al., 2007*); that is, repetitive behavior seems to antagonize social behavior. This antagonistic relationship between social and repetitive behaviors has rarely been reported in other vertebrates. In cavefish, we observed a highly biased one-way turning, based on the ratio of the numbers of left and right turns occurring per fish in each 0.25 s interval, during the 5 min recording (*Figure 4A, C and E*). There was, however, no detectable left-right preference in the direction of turning among individual cavefish (*Figure 4C*). In contrast, surface fish showed symmetrical and balanced turning tendencies (*Figure 4B and D*). As with cavefish, there was no significant left or right preference in the turning direction in surface fish (*Figure 4B*). The levels of bias and direction preference were consistent for each individual cavefish over at least 10 days, indicating that biased

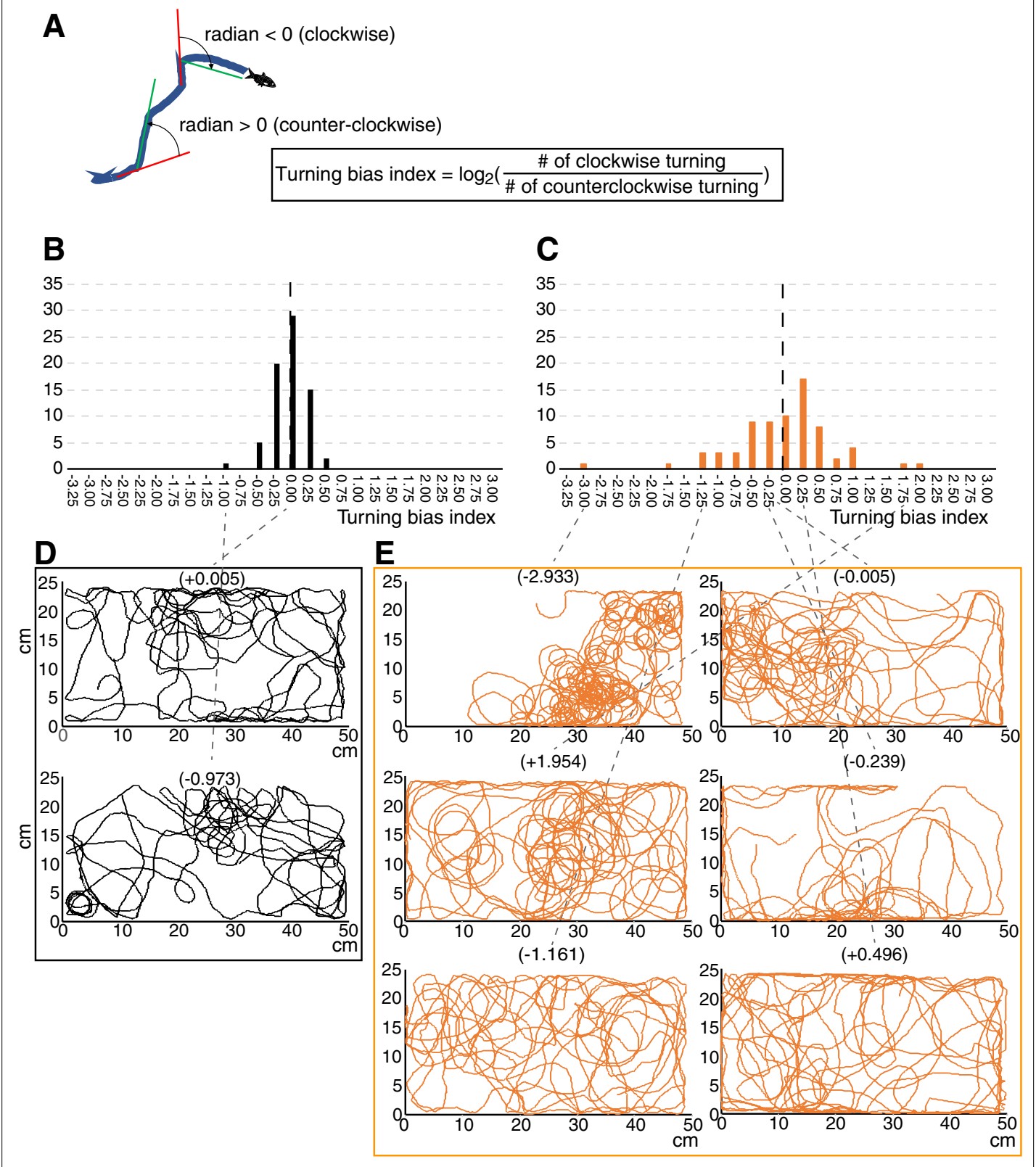

**Figure 4.** Turning bias index showed strong biased turning in cavefish. (**A**) Diagram showing the calculation formula for the turning bias index. The changes in travelling directions were calculated every five frames (every 0.25 s) across all trajectories and were expressed as radians. Positive radian values represent left (anticlockwise) turning, and vice versa. Then, the $\log_2$ of the ratio between the numbers of clockwise and anticlockwise turns was used as the turning bias index. (**B, C**) Histograms of the turning bias index for 72 surface fish (**B**) and 72 cavefish (**C**) in the unfamiliar environment (see

*Figure 4 continued*

the main text). (**D, E**) Swimming trajectories representing a range of turning bias indices in surface fish (**D**) and cavefish (**E**). The replication test regarding turning bias in cavefish is shown in *Figure 4—figure supplement 1*. The source data for the histogram are available in *Figure 4—source data 1*.

The online version of this article includes the following figure supplement(s) for figure 4:

**Source data 1.** The turning indices of surface fish and cavefish.

**Figure supplement 1.** Replication test assessing turning bias in cavefish.

**Figure supplement 1—source data 1.** Repeated measurement of turning indices.

turning is a stable phenotype (interclass correlation: kappa = 0.92, p = 0.0016, N = 6) (*Shrout and Fleiss, 1979*; *Figure 4—figure supplement 1*). Using this biased turning index, we analysed its relationship with the new unfamiliar environment and nearby interactions.

In mammals, stereotypic repetitive motion is enhanced within novel unfamiliar environments (*Boulter et al., 2014*; *Buhr and Dugas, 2009*). Cavefish showed high levels of repetitive turning behavior in both familiar and unfamiliar environments, where a shift in the level was not detectable (*Figure 3E*, *Supplementary file 1*). Additionally, surface fish showed less repetitive turning in both the familiar and unfamiliar environments (*Figure 3E*, *Supplementary file 1*). We therefore concluded that biased turning is a stable phenotype under these two tested environments.

We then tested whether repetitive turning was associated with reduced social-like interaction in cavefish. We did not include surface fish in this correlation analysis because surface fish showed almost no variation in repetitive turning (*Figure 3E*). By fitting a logarithmic curve, based on a logarithmic conversion in the turning bias index, significant correlations between turning bias and duration and between turning bias and nearby-interaction bout number were detected in cavefish in the familiar environment ($R^2$ = 0.198, p = 0.0038; $R^2$ = 0.204, p = 0.0033, respectively; *Figure 5A and B*). However, in the unfamiliar environment, where the nearby interactions were reduced in cavefish, no or less significant correlations between nearby interactions and turning bias was detected ($R^2$ = 0.069, p = 0.0659 and $R^2$ = 0.094, p = 0.0384, respectively; *Figure 5C and D*), possibly due to the significantly reduced variation in nearby interactions.

In summary, a negative correlation between social-like nearby interactions and restricted repetitive behavior was revealed in cavefish under non-stressful conditions. This relationship is shared between mammals and cavefish (*Boulter et al., 2014*; *Buhr and Dugas, 2009*; *Kim et al., 2016* personal communication with Dr. Ryan Lee). Additionally, the reduction in nearby interactions in an unfamiliar environment in cavefish was similar to that in mammals (*Boulter et al., 2014*; *Buhr and Dugas, 2009*; *Kim et al., 2016*), suggesting the possibility of shared neural processing between fish and mammals in these behavioral expressions.

## Involvement of the dopaminergic system in the balance between levels of nearby interactions and turning bias

In mammals, one of the brain regions involved in the expression of restricted repetitive behavior is the striatum (*Kim et al., 2016*; *Langen et al., 2011b*; *Lewis et al., 2007*). The cortico-striatal system is thought to be deeply conserved from teleosts to mammals, and dopaminergic inputs from the ventral tegmental area (VTA) and substantia nigra pars compacta (SNpc) play an important role in behavioral choice in this system (*Grillner and Robertson, 2016*; *Kim et al., 2016*; *Stephenson-Jones et al., 2011*). Within the striatum, the balance between the D1 direct and D2 indirect pathways is crucial, from lamprey to mammals, in balancing behaviors (*Grillner and Robertson, 2016*; *Kim et al., 2016*; *Stephenson-Jones et al., 2011*). An antipsychotic drug, aripiprazole, is one of two approved drugs for ASD treatment by the U.S. Food and Drug Administration (FDA). Aripiprazole suppresses the phasic stimulation-dependent activity of the indirect D2 pathway but preserves the basal tonic level of the D2 indirect pathway due to its partial antagonistic action on the D2 receptor in mammals (*de Bartolomeis et al., 2015*). Aripiprazole is prescribed to reduce irritability, perhaps because it maintains the basal inhibitory activities of the D2 pathway and reduces repetitive behaviors (*de Bartolomeis et al., 2015*). To address whether there is a parallel response to this drug between *A. mexicanus* and humans, we treated surface fish and cavefish with aripiprazole through bath application and tested whether the treatment affected social-like nearby interactions and repetitive turning in both surface fish and cavefish.

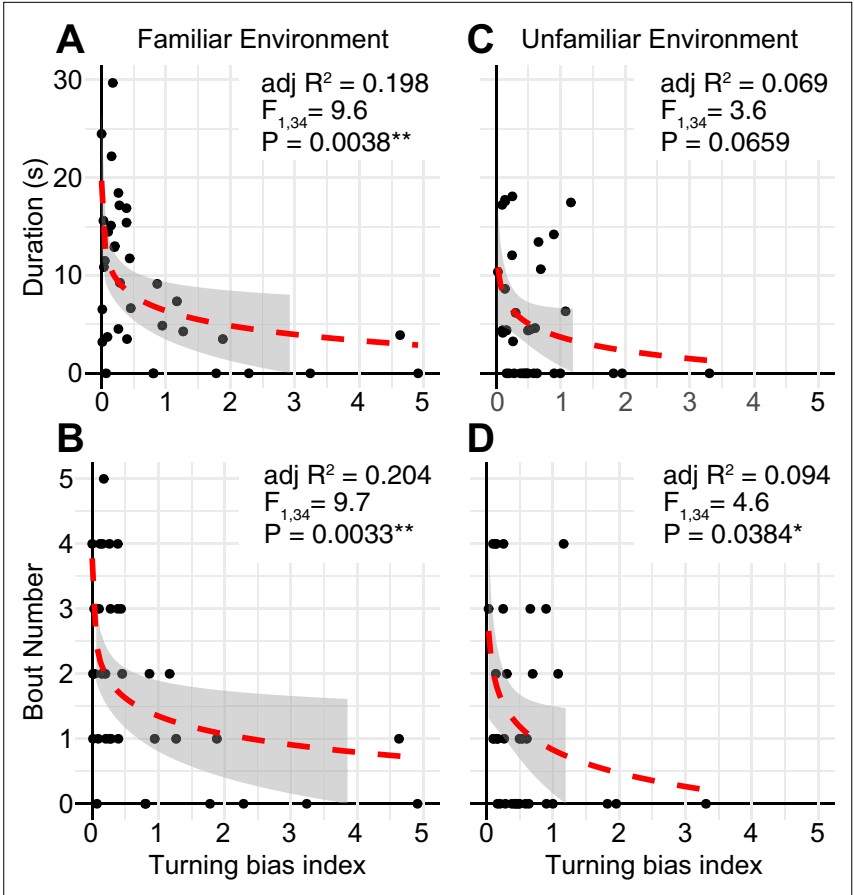

**Figure 5.** The relationship between repetitive turning behavior and nearby social-like behavior in cavefish. (**A, B**) In the familiar environment, the logarithmic transformed turning bias index and aspects of nearby interactions were significantly negatively correlated, $F_{1,34}$ = 9.6, adjusted $R^2$ = 0.198, p = 0.0038 for the turning bias index against the nearby-interaction durations (**A**); $F_{1,34}$ = 10.0, adjusted $R^2$ = 0.204, p = 0.0033 for the turning bias index against the number of nearby-interaction bouts (**B**). (**C, D**) In contrast, in the unfamiliar environment, aspects of the nearby interactions were reduced, and lower correlations were detected, $F_{1,34}$ = 3.6, adjusted $R^2$ = 0.069, p = 0.0659 for the turning bias index against the nearby-interaction durations (**C**); $F_{1,34}$ = 4.6, adjusted $R^2$ = 0.094, p = 0.0384 for the turning bias index against the number of nearby-interaction bouts (**D**). *: p < 0.05, **: p < 0.01. Red dashed lines: regression lines with the logarithmic transformation on the x-axis. Gray shades represent 95 % confidence intervals of the regression lines (N = 36 from nine groups).

The online version of this article includes the following figure supplement(s) for figure 5:

**Source data 1.** The turning bias indices and nearby interaction under the familiar or unfamiliar environment.

Consistent with our previous findings (*Yoshizawa et al., 2018*), both surface fish and cavefish had lower overall swimming distance travelled following aripiprazole treatment compared to the control or pre-treatment levels ($X^2$(1) = 17.4, p = 3.05 × 10⁻⁵; *Figure 6A*, *Supplementary file 1*). Following this treatment, cavefish showed reduced repetitive biased turning (W = 642, p = 0.0170; *Figure 6B*, *Supplementary file 1*) and increased social-like nearby interactions (W = 279, p = 0.0375; *Figure 6C*, *Supplementary file 1*). Surface fish did not show changes in either repetitive biased turning or social-like nearby interactions (p > 0.05; *Figure 6B, C and D*; *Supplementary file 1*). The simulated random data derived from the X-Y coordinates of aripiprazole-treated fish also supported that the nearby interaction durations in both surface fish and cavefish were significantly longer than those expected by chance (*Figure 6—figure supplement 1*). Aripiprazole treatment did not change the profile of velocity shifts during the bout in either surface fish or cavefish: surface fish did not reduce the swimming speed during bouts; in contrast, cavefish reduced swimming speed during bout compared with other periods (surface fish PrePost × Treat × Speed: $X^2$(3) = 0.4, p = 0.9431; cavefish PrePost × Treat × Speed: $F_{3,140}$ = 1.5, p = 0.2073; *Figure 6E and F*, *Supplementary file 1*).

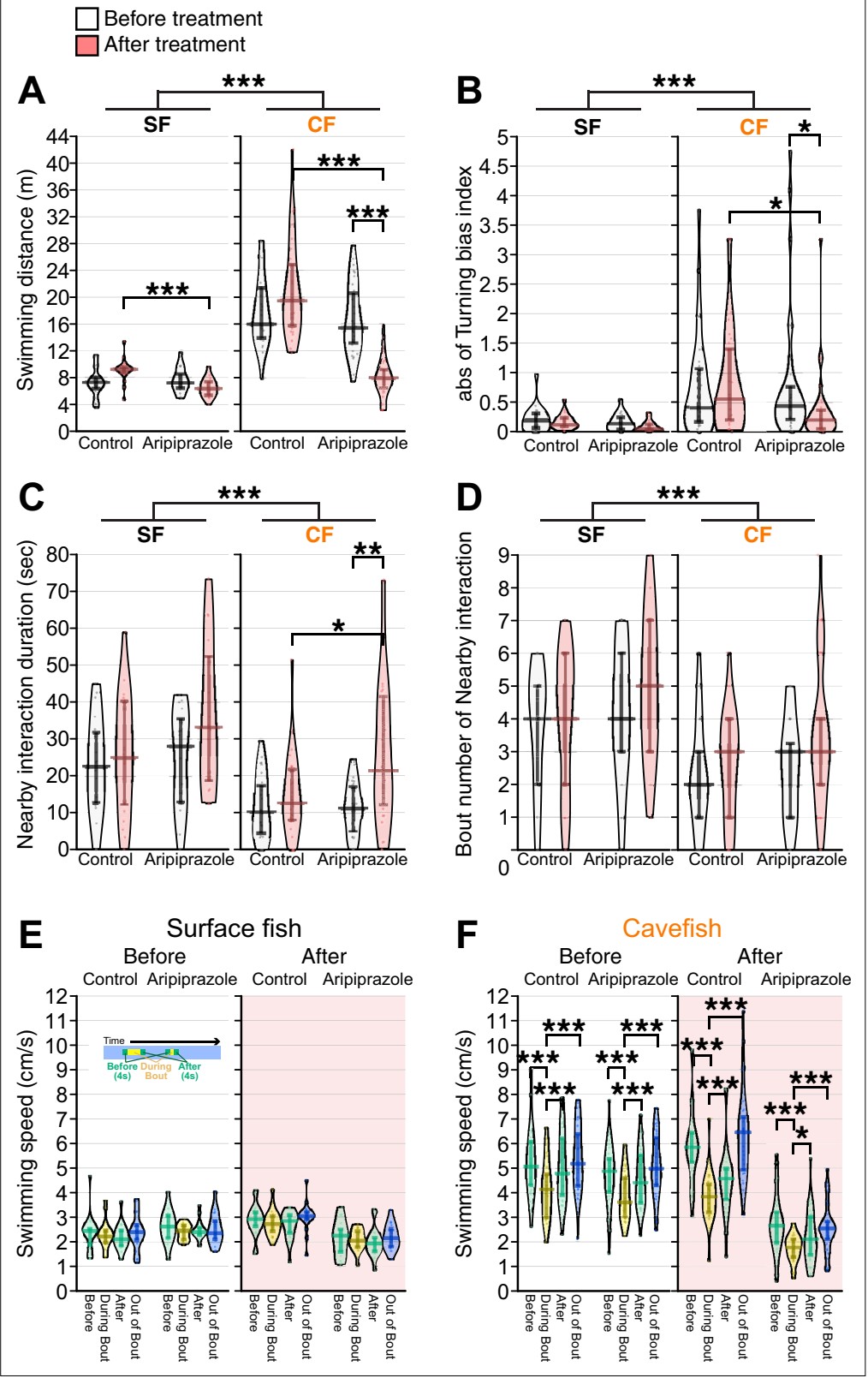

**Figure 6.** The partial agonist of the D2 receptor aripiprazole enhanced nearby interactions and reduced turning bias in cavefish. The pre-treatment behaviors were recorded for more than four days before the aripiprazole treatment. Aripiprazole and the control DMSO were administered for ~16 hr before the post-treatment recording in an unfamiliar environment. (**A**) Total swimming distances before (black and white) and after (red shaded) the

*Figure 6 continued on next page*

*Figure 6 continued*

treatment, between the aripiprazole and control treatments, and between surface fish and cavefish. The cavefish had significantly reduced swimming distances after aripiprazole treatment. (**B**) The turning bias index indicated that cavefish significantly reduced biased turning after aripiprazole treatment. (**C**) The nearby-interaction durations showed a significant difference in Pop (surface fish and cavefish) and a significant interaction with PrePost (pre- and post-treatment)× Pop via a general linearized model (*Supplementary file 1*). Post hoc tests indicated a significant increase in nearby interactions in the aripiprazole-treated cavefish. (**D**) Significant difference in Pop (surface fish and cavefish). (**E, F**) Swimming speed during bouts or out-of-bout periods in surface fish (**E**) and cavefish (**F**). Aripiprazole treatment did not change the speed profiles in either the surface fish or cavefish. N = 20 from five groups for surface fish control; N = 16 from four groups for aripiprazole-treated surface fish; N = 28 from seven groups for cavefish control; N = 32 from eight groups for aripiprazole-treated cavefish. *: p < 0.05, **: p < 0.01, ***: p < 0.001. All statistical scores are available in *Supplementary file 1*. Regression plots between the nearby-interaction durations and swimming speeds are available in *Figure 6—figure supplement 1*.

The online version of this article includes the following figure supplement(s) for figure 6:

**Source data 1.** The swimming distances, absolute values of the turning indices, nearby interaction durations, nearby interaction event numbers, and swimming speeds under the aripiprazole treatment.

**Figure supplement 1.** Nearby-interaction durations were significantly longer in the actual fish than in the simulated random data in the aripiprazole treatment.

**Figure supplement 1—source data 1.** The swimming speed and nearby interaction duration in cavefish and surface fish.

**Figure supplement 2.** Swimming speed is not a significant predictor of nearby-interaction durations.

**Figure supplement 2—source data 1.** Nearby-interaction durations were significantly longer in the actual fish than in the simulated random data in the aripiprazole treatment.

---

To test whether the overall activity reduction itself is the factor that induced the duration and number of interaction bouts, we performed correlation analyses between swimming speeds and nearby-interaction durations. Swimming speed was not a significant predictor of nearby interactions in either surface fish or cavefish pre- or post-aripiprazole treatment (p > 0.05; *Figure 6—figure supplement 2*). These results suggested that aripiprazole induced nearby interactions in cavefish not only because of reductions in the swimming speed.

## Discussion

In this study, we document that the sighted surface-dwelling form of *Astyanax mexicanus* is able to follow and interact with conspecifics in the dark. We also reveal that so-called 'asocial' cavefish retain a low but detectable level of ability to engage in social-like interactions. The social-like interaction depended on the familiarity to the environment, and interestingly, surface fish and cavefish showed opposite responses to the environments: in the familiar environment, cavefish showed more social-like nearby interactions, while surface fish showed reduced social interaction; in contrast, in the unfamiliar environment, surface fish tended to exhibit a longer duration of nearby interaction, while cavefish exhibited a reduced duration of nearby interaction. Cavefish also showed a significant negative correlation between the levels of repetitive turning behavior and social-like interactions in the familiar environment. This quantitative antagonistic relationship between repetitive behavior and social-like interactions has been reported in studies involving humans and rodents (*Boulter et al., 2014*; *Buhr and Dugas, 2009*; *Hong et al., 2014*), but there are few reports in teleosts. Treatment with the partial D2 receptor antagonist aripiprazole enhanced social-like interactions and suppressed repetitive turning in cavefish, which is consistent with aripiprazole treatment results in human patients (*de Bartolomeis et al., 2015*; *Ghanizadeh et al., 2014*; *Nasrallah, 2008*).

In terms of measuring collective behaviors, the current definition of nearby interaction is close to that of 'attraction' in zebrafish larvae under lighted conditions (*Hinz and De Polavieja, 2017*) (for a more general definition, see *Lukeman et al., 2010*), and we believe these two indices indicate the same interactions. However, we could not apply Hinz's criteria to ours because they used larval fish (~24 days post-fertilization: 0.6 cm in body length; *Parichy et al., 2009*). Larval fish swim in a highly viscous low Reynolds number-fluid environment (*Vogel, 1983*) where the larval fish do not glide. Thus, the turning direction that Hinz et al. measured was sufficient to measure collective behavior. In

contrast, we used young adult fish (1.5–2.0 cm), who swam in a higher Reynolds number-fluid environment and glided while swimming. Therefore, this measure of nearby duration that results from the manoeuvring efforts of the fish was more appropriate for representing the level of collective behavior than simply measuring the turning direction.

As for a potential sampling bias in this study, we equally mixed age-matched individuals from different parents. We did not exclude any outliers from our dataset except those showing atypical pre-treatment behaviors (see Materials and methods). Although our lab experienced a founder bottleneck started with approximately 80 individuals per population, we minimized potential sampling bias within our lab populations by the above procedure.

The majority of studies on shoaling and schooling behaviors of teleost fish have been conducted under environmental conditions where visual cues predominate, potentially exaggerating the importance of the visual system in these behaviours (*Larsch and Baier, 2018*). Observations of the maintenance of shoaling behavior in blinded fishes support the involvement of lateral line mechanosensing; however, few studies have used intact fish to test the involvement of the lateral line (*Partridge and Pitcher, 1980*; *Pitcher, 1979*; *Pitcher et al., 1976*). Furthermore, many shoaling tests have been conducted in somewhat physically constrained conditions (e.g. using a transparent glass divider between stimulus fish and test fish [*Dreosti et al., 2015*] or using a small mesh cage that constrains the swimming area of the stimulus fish [*Gerlai, 2014*]), where the true extent of mechanosensing in the control of shoaling behavior may be underplayed. In this study, we conducted the first systemic analysis of social-like nearby interactions in intact fish under free-swimming conditions in the dark.

In this study, we carefully set up and tested whether sighted surface fish or blind cavefish can respond to visual stimuli by measuring phototaxis and monitoring the response to moving object shade (inserted between the 850 nm infrared light source and fish arena). Our results indicated that neither fish responded to these visual stimuli at a detectable level. Although we cannot exclude the possibility that these fish may still receive low levels of visual stimuli, they do not elicit major behavioral outcomes under the current setup.

Using our method of measuring nearby interactions via idTracker-based tracking of each individual, a 'nearby interaction' was defined by both a threshold maximum proximity distance between two fish ($\leq 5$ cm) and a minimum threshold interaction duration ($\geq 4$ s). This method was sensitive enough to detect changes in nearby interactions in different environments (e.g. familiar and unfamiliar; *Figure 3*) and following drug treatment (*Figure 6*; *Figure 2—figure supplement 4*). This method is also applicable to measure a unit of collective behavior among larger groups (i.e. two fish attractions or nearby interactions among ≥5 individuals) in other animal species.

Regarding non-visual collective behavior in *A. mexicanus*, a quantitative trait locus (QTL) for non-visual schooling has been reported where $F_2$ hybrid fish generated from a pair of a surface fish and a Tinaja cavefish (a cave population that evolved independently from that of the Pachón cave; Pachón cavefish were used in this study) were used (*Kowalko et al., 2013*). The schooling measurement was based on how the $F_2$ individual responded to the school of 'model fish' and based on NND. The major schooling QTL is associated with vision (*Kowalko et al., 2013*). The QTL for non-visual schooling was detected by using only $F_2$, which showed negative phototaxis (i.e. had visual capacity) but exhibited large differences from non-schooling to schooling. The method presented here can improve the detection resolution in terms of nearby interactions. Regardless, it was interesting that $F_2$ individuals carrying homozygous cave genotypes at a marker underlying this QTL exhibited increased schooling behavior, while $F_2$ individuals with homozygous surface or heterozygous genotypes schooled less (*Kowalko et al., 2013*). This genetic study suggests two possibilities: (1) Tinaja cavefish evolved a 'schooling allele', which is different from the 'nearby interaction alleles' that Pachón cavefish have, or (2) the same allele regulates Tinaja's schooling behavior and Pachón's nearby interaction. Note that Tinaja's schooling allele may simply be an appetite-driving gene since aggression or foraging against fish plastic models was not evaluated in the former study (*Kowalko et al., 2013*), and Tinaja cavefish show insatiable appetites (*Aspiras et al., 2015*) (see below for cannibalistic foraging). Based on this complexity, a crossing experiment between Tinaja and Pachón cavefish is necessary to answer these possibilities.

Based on our results of pharmacological ablation of the lateral line, the nearby interaction of surface fish in the dark is likely regulated by lateral line mechanosensors (*Figure 2—figure supplement 4*). However, this result does not mean that the involvement of other sensory systems, such as

tactile and olfactory systems, can be excluded (*Timmermann et al., 2004*). Further tests are needed to address which of these sensory systems plays the crucial role in regulating nearby interactions in fish in the dark.

In the dark, surface fish changed their swimming direction based on the presence of other individuals, as if they sensed and were attracted to others, similar to what has been observed in zebrafish larvae under lighted conditions (*Figure 2—figure supplement 2*; *Hinz and De Polavieja, 2017*). However, surface fish failed to form a clear cluster, that is a 'shoal', in the dark. Instead, surface fish individually responded to one another. We believe that the failure to form a shoaling cluster was based on the lack of visual information (see *Videos 1 and 4* under light and dark conditions, respectively), yet surface fish maintained their collective tendency in the dark by using lateral line sensing.

The nearby interactions detected in this study did not appear to be related to foraging activities or aggression (i.e. two fish come close while expressing aggression or cannibalistic foraging [*Elipot et al., 2013*]). We categorized the nearby activities into five classes of activity and visualized them to show the frequent role changes of leaders and followers within single bouts in the same fish pair (*Figure 2—figure supplement 2C and E*). This role change is rarely observed in aggressive encounters (*Elipot et al., 2013*). In addition, cavefish slowed during the nearby-interaction bout compared with other periods of time, which contradicts the speed shift observed during aggression—fish thrust during aggression (*Elipot et al., 2013*). Finally, we carefully assessed damage to the fins and scales, and although these tissues are good indicators of aggressive encounters, noticeable damage was not detected. Consistent with this observation, both surface fish and cavefish rarely showed startle responses after touching one another. We therefore conclude that *A. mexicanus* rarely attacked one another in our assay system and that the majority of detected nearby interactions can be characterized as shoaling-like activities.

The degree and type of nearby interactions in the dark that were recorded in this study appear relevant within the context of each morph's natural habitat. Surface fish have a known nocturnal predator, prawn fish (*Wilson et al., 2004*), and adult surface fish form a tighter shoaling cluster at night (personal observation in the laboratory). These facts imply that shoaling can be advantageous by simply lowering individual risk among cohorts, the so-called 'attack dilution effect', and/or by increasing sensitivity in detecting potential predators by the collective effect of more sensors of multiple individuals (*Pitcher and Parrish, 1993*). In cavefish, a higher level of nearby interactions detected in the familiar laboratory setting was also observed in their natural habitat, Cueva de El Pachón, Tamaulipas, Mexico (2019, personal observation). While they did not form a recognizable shoal, cavefish responded to conspecifics and tended to follow each other. This social-like tendency could merely be a residual of regressive evolution to a troglobitic niche following their divergence from surface riverine ancestors (*Langecker et al., 1991*; *Yoshizawa et al., 2012*). Alternatively, this weak social-like tendency could remain advantageous for finding mates, locating food or evading potential dangers within their cave environment.

The young adult surface fish used in this study had reached an appropriate developmental stage to express the shoaling behavior, and all of their behaviors were assessed only in the day. Since the shoaling motivation was seen at night with very limited visual inputs (see above), it is important to investigate shoaling capacity at night in future studies.

In the unfamiliar setting, the cavefish reduced the social-like interaction and kept high-level of repetitive turning, suggesting that cavefish lost the strategy to use social-like interactions to confront unfamiliar environmental inputs, which surface fish have.

Restricted repetitive behaviors are observed in mammals kept in restricted/isolated environments and in humans with stereotypic movement disorders, including ASD (*Lewis et al., 2007*; *Rose et al., 2017*). This motor disorder has been considered non-functional. However, since these stereotypic behaviors are seen in many vertebrate species as well as in typically developing human infants, there may be an advantage to repetitive behaviours. To understand the behavioural function of repetitive motion, an emerging hypothesis—predictive processing—was introduced in cognitive neuroscience (*Palmer et al., 2017*). In this framework, to develop an internal model for the habitat ( = environment), the central nervous system is continuously attempting to minimize prediction errors, that is, differences between the environmental signals sampled by an animal's sensory systems and that animal's internal model of the world within the brain. The brain is assumed to have two strategies to minimize differences between sensory input and the internal model when novel stimuli are encountered: (i) to

use a better fitting model among many existing internal models or (ii) to locomote back to a 'no-novel stimulus environment' to sample the sensory inputs that had already been implemented in the current internal model (*Friston, 2010*). Stereotypic repetitive circling seems to be a result of choosing the latter strategy by sampling predictable sensory inputs (strong lateral line inputs on one side of the body). This method can reduce the prediction errors between the sensory inputs and those predictions from the internal model (e.g. left turning provides the strong right lateral line signals in cavefish). In contrast, surface fish responded to unfamiliar stimuli and did not increase repetitive turning, perhaps by selecting a better fitting model among the available internal models (i.e. strategy (i)). Therefore, based on this predictive processing framework, surface fish and cavefish apparently have different strategies. Both of these changes could minimize the errors between the sensory inputs and the predicted inputs associated with the internal model. Given that surface fish appear to acclimatize to novel environments much quicker than cavefish measured by foraging latency (personal observation), this suggests that surface fish may store many internal models that are proxies of the current environment. Having many internal models is thought to better highlight new salient stimuli in a changing environment (i.e. a good fitting-internal model yields better attention to the new object). The difference in the richness of internal models between surface fish and cavefish is addressable by examining relationships between a number of different environmental setups and how individuals can accurately navigate to reach food without panicking/repetitive motion.

Previous studies have indicated that there is an antagonistic relationship between repetitive and social behavior (*Kim et al., 2016*; *Lewis et al., 2007*). For example, social impairment and repetitive behavior are thought to be linked together as revealed by results following oxytocin infusion (reviewed in *Kim et al., 2016*). In contrast, in a study of ketogenic diet treatment in individuals with autism, social behavior in some treated patients had recovered to normative levels, while repetitive behaviour was not significantly attenuated (*Lee et al., 2018*). This result suggested that repetitive behavior can be independent of social impairment. The mechanisms that controls switching between social and repetitive behaviors is still largely unknown. Many studies of repetitive behavior and social behavior have focused on the cortico-striatal system, where many patients who show repetitive behavior (including those with Gilles de la Tourette syndrome, obsessive–compulsive disorder, ASDs, Parkinson's disease, and Huntington's disease) have abnormal functional and/or histological features diagnosed by PET and fMRI (*Langen et al., 2011a*). Importantly, this striatal system, particularly its dopaminergic system, has also been suggested to drive social behavior, where the dopamine signal from the VTA stimulates D1-dopamine receptor-positive striatal neurons in the nucleus accumbens, driving social behaviour in mice (*Gunaydin et al., 2014*; *Kim et al., 2016*). Among vertebrates, the striatum has been proposed as the conserved bridging centre. The striatum integrates (1) higher levels of sensory inputs from the cerebral cortex/dorsal pallium and (2) information-evaluating dopaminergic input from the substantia nigra par compacta (SNc). The dopaminergic striatal pathway is largely conserved among vertebrate species. This includes the direct and indirect pathways, each of which is associated with a specific type of dopamine receptor—D1 and D2 receptors, respectively. The major function of this striatal dopaminergic system is thought to be to select behavioral modules (e.g. mating, foraging, and aggression modules) based on given sensory inputs and internal states (*Grillner, 2018*; *Grillner and Robertson, 2016*). For example, a fish may ignore or follow mechanical signals from cohorts based on the fish's internal state.

In addition to the striatal behavior switching centre, many other brain regions are known to induce social/social-like behaviors. For example, from teleosts to mammals, the best neurochemically conserved social decision-making network has been reported to be the preoptic area (POA based on an analysis of 10 gene products across 12 brain decision-making processing regions in 88 species; dopaminergic neurons also exist in this region) (*O'Connell and Hofmann, 2012*; *O'Connell and Hofmann, 2011*).

Accordingly, bath treatment with a partial D2-dopamine receptor antagonist, aripiprazole, showed a parallel response between cavefish and humans, suggesting a potential shared pathway that controls social-like and repetitive behaviors in teleosts and mammals and possibly involves the striatum, POA, and/or other dopaminergic processing nuclei (*Grillner, 2018*; *Kim et al., 2016*; *O'Connell and Hofmann, 2011*). Future studies will address whether the antagonistic relationship between social-like and repetitive behaviors is based on the switch of behavior modules in the striatal system or whether repetitive motions simply fragment social-like interactions, although the individual may continue to have the motivation to congregate.

It will be exciting to examine the evolutionary and neurophysiological changes in the dopaminergic system between cavefish and surface fish and how these changes are associated with the integration of an internal model, which the predictive-processing theory proposes, in the future.

# Materials and methods

**Key resources table**

| Reagent type (species) or resource | Designation | Source or reference | Identifiers | Additional information |
|---|---|---|---|---|
| Strain, strain background (*Astyanax mexicanus*) | Surface (epigean) population | Yoshizawa lab descendant of William R Jeffery lab at Univ Maryland College Park US | SFe; SFg | DOI:10.1126/science.289.5479.631 |
| Strain, strain background (*Astyanax mexicanus*) | Cave (hypogean) population | Yoshizawa lab descendant of Richard Borowsky lab at New York Univ. US | PAby | DOI:10.1101/pdb.prot5093 |
| Chemical compound, drug | Aripiprazole | Sellek Chemicals | S1975 | |
| Chemical compound, drug | Gentamicin | MilliporeSigma | G1914 | |
| Software, algorithm | MATLAB | MathWorks | R2020a | |
| Software, algorithm | Python | Python | Python 3.7.4 | |
| Software, algorithm | idTracker | *Pérez-Escudero et al., 2014* | idTracker-beta | https://github.com/idTracker/idTracker/tree/Beta |
| Chemical compound, drug | 4-Di-2-Asp | MilliporeSigma | D3418 | |

## Fish maintenance and rearing in the laboratory

The *Astyanax mexicanus* surface fish used in this study were laboratory-raised descendants of original collections made in Balmorhea Springs State Park, Texas, by Dr. William R. Jeffery. The cavefish were laboratory-raised descendants originally collected from Cueva de El Pachón in Tamaulipas, Mexico by Dr. Richard Borowsky.

Fish (surface fish and Pachón cave populations) were housed at the University of Hawaiʻi at the Mānoa aquatic facility with temperatures set at 21°C ± 0.5°C for rearing, 22°C ± 0.5°C for behavioral experiments, and 25°C ± 0.5°C for breeding (*Elipot et al., 2014*; *Yoshizawa, 2015*). Lights were maintained on 12:12 light:dark cycles (*Elipot et al., 2014*; *Yoshizawa, 2015*). For rearing and behavioral experiments, light intensity was maintained between 30–100 lux. Fish were raised to 7–12 month-old young adults and maintained in plastic containers (Ziplock containers, S. C. Johnson & Son, Inc, Racine, WI, USA) with ~20 cohorts. Young adult fish were fed live brine shrimp nauplii to satiation two times daily starting 3 hr after the lights came on (Zeitgeber time three or ZT3) and ZT9 (Premium Grade Brine Shrimp Eggs, Brine Shrimp Direct, Ogden, UT, USA). All fish in the behavioral experiments were between 3 and 4 cm in standard length and between 7 and 12 months old. Regarding the selection of experimental fish, as often as possible, all the fish groups in our study comprised age-matched siblings. We then equally mixed age-matched individuals from different parents within the morph (cavefish or surface fish). We did not exclude any outliers from our dataset except those showing atypical pre-treatment behaviors, such as being still at the bottom for more than 3 min or showing continuous repetitive circling of more than 5 s in their home tanks. All fish care and experimental protocols were approved by the IACUC (17–2560) at the University of Hawaiʻi at Mānoa. All populations of *A. mexicanus* used in this study are readily available from us upon request.

## Recording fish group behavior

Prior to experimentation, four fish comprising group-raised siblings were housed in a 'home' container (15.6 × 15.6 × 5.7 cm: Ziplock containers, S. C. Johnson & Son, Inc) filled with conditioned water (pH 6.7–7.3; conductivity 600–800 µS) for 4 or more days. Fish were fed twice a day with live brine shrimp nauplii until the morning of the recording. For the unfamiliar environment experiment, all four fish were gently transferred to a 37.85 L tank (50.8 × 25.4 × 30.5 cm) with a water depth of 2 cm on the stage of a custom-made infrared (IR) back-light system within a custom-built black box (75 × 50 × 155 cm, assembled with polyvinyl chloride (PVC) pipe frame and covered by shading film). The IR back-light system was composed of bounce lighting of IR LED strips (SMD3528 850 nm strip: LightingWill, Guang Dong, China) (*Figure 1A*). Videos were recorded using a USB webcam (LifeCam Studio 1080 p

HD Webcam, Microsoft, Redmond, WA, USA) fitted with a c-mount zoom lens (macro 1.8/12.5–75 mm C-mount zoom lens, Toyo Lens, Tokyo, Japan) installed on the top of the custom-built black box. An IR high-pass filter (Optical cast plastic IR long-pass filter, Edmund Optics Worldwide, Barrington, NJ, USA) was placed between the camera and the lens to block visible light. The video was captured at 20 frames per second using VirtualDub software (version 1.10.4, http://www.virtualdub.org/) or VirtualDub2 (build 44282, https://sourceforge.net/projects/vdfiltermod/) with a vfw codec (https://sourceforge.net/projects/x264vfw/). After 6 min of recording, the fish were returned to the housing container. For the familiar environment experiment, fish were transferred to 37.85 L tank (50.8 × 25.4 × 30.5 cm) with a water depth of 2 cm and kept for 4 days to let them acclimate. On the fourth day, prior to the recording, we monitored the water qualities of each tank (pH, conductivity, and ammonia, nitrite, and nitrate levels) and then performed the recording in tanks with healthy water conditions (i.e. pH = 6.7–7.3, conductivity = 680–800 μS, almost 0 in ammonia, nitrite and nitrate levels). All tanks met these criteria. The movements of the acclimated fish were recorded using the same acclimation tank and water. After all recording sessions, the study fish were visually checked for damage to the fins and for lost scales as an indicator of overt aggression.

## Drug treatment

The bath treatment of aripiprazole (Selleck, Houston, TX) was performed as described in *Yoshizawa et al., 2018*. Briefly, the aripiprazole working solution was prepared by mixing 1.0 mM aripiprazole stock solution into fish-conditioned water to make a 1000× dilution (final concentration: 1.0 μM aripiprazole and 0.01 % DMSO in the conditioned water). For the stock solution, aripiprazole (Selleck, Houston, TX) was dissolved in 100 % dimethyl sulfoxide (DMSO, Millipore Sigma). For the control, 0.01 % DMSO in conditioned water was used. The four fish in each home container were treated with either the aripiprazole working solution or the 0.01 % DMSO control solution overnight (~16 hr). The next morning, the fish group behaviors in the 37.85 L tank were recorded as described in the 'Recording fish group behavior' section.

## Tracking

The first minute of each original video was discarded since this time interval contains the moment that four fish were released from their home container, which agitated the water surface, disrupting observation. Before tracking each fish's position by idTracker software (*Pérez-Escudero et al., 2014*), the background visual artefacts were subtracted by a custom-made ImageJ macro script. Briefly, with videos recorded with a white background, each frame image was adjusted by the 'Add' operation of the 'Image Calculator' function with an inverted median z-stacked image of 2000 frames calculated by 'Z Project...' and 'Invert' functions of ImageJ (*Schneider et al., 2012*). Then, the frame images' contrasts were adjusted based on each video. Each processed video was saved in AVI format by using jpeg compression in ImageJ. Using these background-subtracted videos, the X-Y coordinates of each fish were extracted by tracking each fish's ID under the ID-detection algorithm of idTracker (idTracker-beta running under MATLAB release 2019a or above; https://github.com/idTracker/idTracker/tree/Beta; *Pérez-Escudero et al., 2014*; The MathWorks Inc, Natick, MA, USA). The parameters of idTracker were usually set as follows: number of individuals, 4; intensity threshold, 0.75–0.89; minimum size, 50; 'select region by polygon' function enabled; and 'Remove background' option checked. We aimed to achieve a successful tracking frame ratio above 97 % by adjusting these parameters. Even with a high ratio of successful tracking frames (e.g. 99%), the outputs of idTracker can contain ID errors by switching the IDs between different fish. These errors were manually surveyed frame-by-frame by using idPlayer, an accompaniment to idTracker software (*Pérez-Escudero et al., 2014*). The corrected IDs were re-assigned by a custom Python script (https://zenodo.org/record/5122894#.YPnDBR1ujsF). If a fish had periods without an ID during the analysed videos, the estimated X-Y coordinates were calculated for those durations by averaging velocity and swimming direction between the X-Y coordinates before and after the missing ID period. Tracking results were used for the repetitive turning and nearby-interaction analyses (see below).

## Standardization method for nearby interaction in the dark

To characterize social-like interactions, we developed an assay system incorporating a naturalistic free swimming set up that also deprived the fish of visual stimulus (see body text too). In previous reports,

teleost fish display a wide array of collective behaviors (e.g. schooling and shoaling)(**Pitcher and Parrish, 1993**). Attempts have been made to quantitatively assay social interactions in some marine (sea beam, salmon) and a few freshwater teleost (cave barb and cave molly and their surface species/populations), revealing the major involvement of visual- and mechanosensing in teleost collective behavior (**Bleckmann, 1986**; **Paciorek and Mcrobert, 2012**; **Pitcher, 1993**; **Pitcher and Parrish, 1993**; **Timmermann et al., 2004**). However, the majority of these reports measured shoaling activity under light conditions (thus under visual stimuli) and by using two-choice setup where a social-target fish was constrained in a small compartment and let the focal fish approach the nearby area for the target fish (**Gerlai, 2014**). The 'constrained' setup is comparable to the popular three-chamber test used in mice to assay sociability and social novelty (**Silverman et al., 2010**). However, this spatial restriction is an unusual stimulus to fish, and does not allow the same pattern of water fluctuation/vortices to form as when fish swim freely, which significantly limits the ability of a focal fish to follow another by using lateral line mechanosensors (**Bleckmann et al., 1991**; **McHenry and Liao, 2013**). The lighted condition also promoted tighter shoaling in surface fish yet no detectable change in cavefish of *A. mexicanus* (**Kowalko et al., 2013**; **Patch et al., 2020**; **Pierre et al., 2020**). Our free-swimming setup is to address nearby interaction with an aid of idTracker auto-tracking software (**Pérez-Escudero et al., 2014**). Under our free-swimming assay system, sighted surface fish tightly shoaled/schooled under lighted condition as reported (**Video 4**; **Keene et al., 2016**; **Kowalko et al., 2013**; **Patch et al., 2020**; **Pierre et al., 2020**).

According to how the nearby interactions happened more than chance, we set cut-off distance and duration of nearby interaction as ≤5 cm and ≥4 sec, respectively (see main text). We also standardized the social bouts with the passing-by duration (PbDur). PbDur is defined here as the average duration of a fish passing-by another fish where these fish do not express the nearby interaction. In order to remove the period that resulted from just physical proximity rather than the actual social-like interaction (been 'attracted'), the PbDur were subtracted from each bout of the nearby interaction durations. Derivation of PbDur as follows.

The average duration of passing-by within each 4-fish group (PbDur) was calculated from the average relative swimming velocity ($V_r$) and average traveling distance within 6 cm radius of another fish (Distance).

$V_r$ is calculated based on the scenario that two nearby fish are swimming in the velocity of $V$—an average velocity of four fish in the same group (Materials and methods and *Figure 2—figure supplement 1C*). Given two fish swim in the velocity of $V_1 = V_2 = V$ (see below box), the relative velocity is:

$$|V_r| = 2 \cdot |V| \cdot sin\left(\frac{\theta}{2}\right). \tag{1}$$

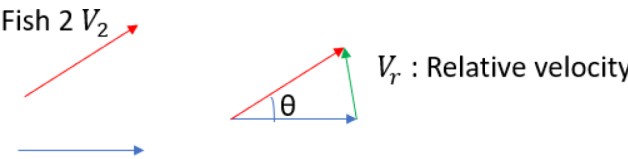

**Scheme 1.** Fish 1 and Fish 2 are swimming in velocities of $V_1$ and $V_2$, respectively. Then, to simplify the calculation, suppose that the two fish swim with the average velocity, $V = V_1 = V_2$, the average of the relative velocity between two fish is given by $V$ and $\theta$ (see above).

Then the average speed of $V_r$ is:

$$E\left(|V_r|\right) = p(\theta) \cdot \int_0^\pi 2 \cdot |V| \cdot sin\left(\frac{\theta}{2}\right) d\theta, \tag{2}$$

where $p(\theta)$ is the uniform probability density distribution between 0 and π and expressed as:

$$p(\theta) = \frac{1}{\pi - 0} = \frac{1}{\pi} \tag{3}$$

Therefore, the average relative velocity becomes:

$$E\left(\left|V_r\right|\right) = \frac{1}{\pi} \cdot 2 \cdot |V| \cdot 2 = |V| \cdot \frac{4}{\pi} \qquad (4)$$

Next, we calculated the average of traveling distances within a 5 cm radius of another fish.

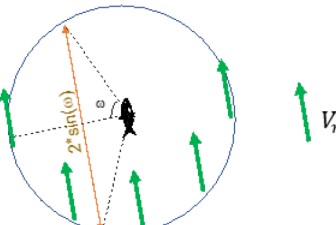

**Scheme 2.** By using relative velocity, we can think as that one fish of a pair is still. Then, while another fish passing by a 5 cm radius of the first fish (circle), this fish travels 5 × 2 × sin($\omega$) cm (orange arrow).

With above assumption, the average of traveling distance $L$ is then:

$$E(|L|) = 5\left(cm\right) \cdot p\left(\omega\right) \cdot \int_0^\pi 2 \cdot sin\left(\omega\right) d\omega, \qquad (5)$$

where

$$p\left(\omega\right) = \frac{1}{\pi - 0} = \frac{1}{\pi} \qquad (6)$$

see (*Equation 3*) too Therefore, the average traveling distance within a 5 cm radius is:

$$E(|L|) = 5\left(cm\right) \cdot \frac{1}{\pi} \cdot 2 \cdot 2 = 5\left(cm\right) \cdot \frac{4}{\pi} \qquad (7)$$

From $E(|V_r|)$ and $E(|L|)$ (*Equation 4* and *7*, respectively), the average passing-by duration (PbDur) is:

$$\text{PbDur} = \frac{E(|L|)}{E(|V_r|)} = \frac{5(cm) \cdot \frac{4}{\pi}}{|V| \cdot \frac{4}{\pi}} = \frac{5(cm)}{|V|}$$

## DASPMI staining of mechanosensory neuromasts

Neuromast vital staining was performed as described previously (*Fernandes et al., 2018*; *Worsham et al., 2019*; *Yoshizawa et al., 2010*). Within 30 min after the video recording, each tested fish was isolated into a 100 ml beaker and immersed in 2.5 µg/ml 4-(4-(dimethylaminostyryl)–1-methylpyridin ium iodide (4-Di-1-ASP or DASPMI; MilliporeSigma, Burlington, MA)) dissolved in conditioned water for 30 min to 1 hr, followed by anesthesia in ice-cold 66.7 µg/mml Ethyl 3-aminobenzoate methane-sulfonate salt (MS-222, MilliporeSigma) in conditioned water. Fish were then visualized under the fluorescence microscope (BX61WI Olympus microscope with 2.5 x MPlanFL N lens, a Rhodamine filter set and an ORCA-Flash4.0 digital camera). After the imaging, fish were recovered in the conditioned water at approximately 22 °C.

## Gentamicin treatment to ablate the lateral line neuromasts

For the gentamicin treatment, gentamicin was bath treated as described (*Yoshizawa et al., 2010*). Briefly, a 0.002 % solution of gentamicin sulphate salt (Millipore Sigma) in conditioned water was prepared and treated for 16–18 hr prior to the experiment. The fish behaviors were assayed as described in the 'Recording group fish behavior' without gentamicin and analyzed as described in the 'Tracking' subsections in the Materials and methods section.

## Biased repetitive turning analyses

Changes in the angles in swimming directions in every five frame windows (i.e. every 0.25 s) were calculated as radians. When the X-Y coordinates of the three time points ($t_0$, $t_1$, and $t_2$) were ($x_0$, $y_0$), ($x_1$, $y_1$) and ($x_2$, $y_2$), the swimming angle ; between two vectors ($x_1 - x_0$, $y_1 - y_0$) and ($x_2 - x_1$, $y_2 - y_1$) was calculated as the cross product of two vectors:

$$\theta = arctan\frac{(x_1 - x_2)(y_0 - y_1) - (y_1 - y_2)(x_0 - x_1)}{(x_1 - x_2)(x_0 - x_1) + (y_1 - y_2)(y_0 - y_1)}.$$

The numbers of frames that had negative radian values ($N_1$), representing clockwise turns, and positive radian values ($N_2$), representing anticlockwise turns, were totalled. The repetitive turning index was then calculated as $log_2 N_1/N_2$ . As fish did not show any preference in the direction of turning (*Figure 4B and C*, and *Figure 4—figure supplement 1*), the repetitive turning index was expressed as the absolute value of $log_2 N_1/N_2$ .

## Recording of single fish repetitive behavior

Repetitive behavior assays were carried out with single fish; each individual fish was, in turn, transferred from the home container to a 25.5 × 38.0 cm Pyrex glass baking dish, filled to a water depth of ~4 cm, by using a fine brine shrimp net. Two of these dish arenas were lit by a custom-made IR light box from underneath (a 58.4 × 41.2 × 15.2 cm opaque storage box equipped with IR LED strip SMD3528 850 nm strip: LightingWill). Videos for this behavioral assay were recorded with the same camera setup as described in the 'Recording fish group behavior' section.

A repeated behavioral assay using the same fish was recorded 10 days after the first. To identify individuals, each fish was labelled with a physically implanted dye (Visible Implant Elastomer (VIE) tags, Northwest Marine Technology, Inc Anacortes, WA, USA) while anaesthetized in a bath treatment of ice-cold 0.2 mg/mL MS-222 (Millipore Sigma, St. Louis, MO), directly after the first recording. After 10 days, the fish behavior was recorded using the same procedure.

## Social-like nearby-interaction analysis

The IIDs and NNDs were calculated according to *Partridge et al., 1980*. For the IID measurement, the average of the distances between the four fish being assayed (six distances) was calculated per video frame. Then, the mean and standard error of means (s.e.m.) were calculated by using one 4-fish group as N = 1 (for example, N = 18 in each of the surface fish and cavefish experimental groups in *Figure 2B*). For the NND measurement, the average of the four nearest distances between a focal fish and the nearest individual per video frame was determined, then the mean and s.e.m. were calculated as with the IIDs (*Figure 2A*).

A 'nearby-interaction bout' was defined by a minimum interaction duration threshold of 4 s and a maximum physical distance threshold of 5 cm distance between the two fish. Notably, the definition of nearby interaction is close to that of 'attraction' by Hinz et al., where attraction was expressed as the probability that the focal turns to the side where the other fish is located (*Hinz and De Polavieja, 2017*). The current nearby interaction measures the preference to stay or follow one another without consideration of the turning direction. We use the term 'nearby interaction' in this study to avoid confusion (see Results section).

Nearby interactions were extracted from each fish's X-Y coordinates by a custom-made MATLAB script. The bout duration was adjusted by subtracting the PbDur (see the following, Materials and methods, and *Figure 2—figure supplement 1C*). PbDur was defined as the average duration of a fish passing by another fish in a situation where these fish do not engage in a nearby interaction. The purpose of subtracting the PbDur from the bout durations was to remove the period from the nearby-interaction events that resulted simply from physical proximity rather than the actual social-like interaction. PbDur was calculated as PbDur = 5 cm $/V$, where $V$ was the average speed of four fish in the same group during the 5 min period (Materials and methods; *Figure 2—figure supplement 1C*). If $V$ is low, PbDur becomes longer; therefore, the removal of the PbDur excludes 'by-chance' events where two fish may remain within 5 cm of one another while not expressing attraction toward one another. Subtraction of PbDur from each detected bout was performed with all presented results including data across surface fish and cavefish, environments, and drug treatments. The swimming velocities 'during the bout', '4 s before bout commencement', and '4 s directly following the bout' were also extracted and calculated using a custom-made MATLAB script.

A random sampling test was performed by random selection of one individual from each group (i.e. one of four fish) and forming a simulated four-fish group, where none of the individuals physically swam in the same arena. We then calculated the IIDs (*Figure 2D*), the duration of each IID event (*Figure 2E*), and the total nearby-interaction duration (*Figure 2G*) among these simulated four-fish groups by using their 5 min X-Y coordinates. The simulated four-fish group was randomly and exclusively formed 1280 times for the tests in *Figure 2D and E* and 18 times for *Figure 2G* by using each surface fish and cavefish dataset.

## Phototaxis and moving object assay

For phototaxis analysis, four fish were released in a half infrared lighted (850 nm: IR LED SMD3528 850 nm strip) and half unlighted arena (*Figure 1—figure supplement 1*) and recorded for 5 min (see Recording fish group behavior). The probability of fish being in the lighted area was calculated per frame ($\frac{number\ of\ fish\ in\ the\ lighted\ area}{4\ fish}$; 5 min ×60 s × 20 frames per second = 6,000 frames; *Figure 1—figure supplement 1*), and then means and 95 % confidence intervals of this probability were calculated across 6,000 frames. For the moving object assay, four fish were released in the arena, and a moving object was introduced between the recording arena and the IR light source (850 nm). Any mechanical noise was minimized while the object was manually moved, such as by avoiding touching the recording arena. Fish and object movements were video recorded for 5 min. The fish behaviour was manually assessed by using the recorded video.

## Statistical analysis

Regarding a power analysis, we designed our experiments based on hree-way repeated measures ANOVA whose effect size was moderate ($f$ = 0.25), with an alpha-error probability = 0.05 and power = 0.80, and the number of groups was 8 (surface fish vs. cavefish× control vs. treated× pre vs. post-treatment). G*Power software (*Faul et al., 2009*; *Faul et al., 2007*) estimated that the sample size needed for this experiment was N = 9 per group. We thus designed to have N ≥ 12 for all experiments in this study.

For statistical comparisons of our data, we performed tests including Student's *t*-tests and Wilcoxon's signed-rank tests and two-way or three-way generalized linear model analyses to compare surface vs. cavefish, treated vs. non-treated, and pre- vs. post-treatments. We calculated Akaike's information criterion (AIC) for each linear and generalized linear model and chose the model with the lowest AIC. The post hoc Holm correction was used to understand which contrasts were significant (*Holm, 1979*).

Regarding replicates of experiments, we used different individuals for the replicates, that is 2 or four biological replicates/trials, by using different individuals in each trial. For example, data from 72 surface fish and 72 cavefish are shown in *Figure 2G and F*. We assayed them over 4 days. On each day, we used four or five groups from each population, and each group included four fish (i.e. (4 + 5 + 4 + 5) × 4 = 72 fish per population). There was no repeated usage of individual fish with the exception of examining familiar and unfamiliar experiments (*Figure 3*). Under these conditions, we confirmed that the averages of each day's data were statistically indistinguishable from each other. We then merged the data acquired on different days and presented the data as a single set of results (i.e. one biological replicate).

The above calculations were conducted using R version 4.0.4 software (packages of car, lme4, and lmerTest) (*Bates et al., 2015*; *Fox and Weisberg, 2019*; *Kuznetsova et al., 2017*), and all statistical scores are available in *Supplementary file 1*, figure legends or the body text.

## Acknowledgements

We thank C Balaan for constructive comments regarding insights on social-like interactions in cavefish. We also thank A Hudson for the English language edit and providing significant improvements in the logical flow of the final draft. We are grateful to V Crystal, J Choi, L Lu, J Nguyen, VFL. Fernandes, K Lactaoen, M Worsham, H Hernandez, N Doeden, J Kato, A Tran, M Ito, R Balmilero-Unciano, E Doy, A Martinez, and D Mones for fish care assistance.

## Additional information

### Funding

| Funder | Grant reference number | Author |
| --- | --- | --- |
| National Institutes of Health | P20GM125508 | Masato Yoshizawa |
| Hawaii Community Foundation | 18CON-90818 | Masato Yoshizawa |

| Funder | Grant reference number | Author |
|--------|------------------------|--------|

The funders had no role in study design, data collection and interpretation, or the decision to submit the work for publication.

## Author contributions
Motoko Iwashita, Conceptualization, Data curation, Formal analysis, Investigation, Methodology, Project administration, Resources, Software, Validation, Visualization, Writing - original draft, Writing - review and editing; Masato Yoshizawa, Conceptualization, Data curation, Formal analysis, Funding acquisition, Investigation, Methodology, Project administration, Resources, Supervision, Validation, Visualization, Writing - review and editing

## Author ORCIDs
Motoko Iwashita http://orcid.org/0000-0002-6653-6823
Masato Yoshizawa http://orcid.org/0000-0001-8455-8252

## Ethics
This study was performed in strict accordance with the recommendations in the Guide for the Care and Use of Laboratory Animals of the National Institutes of Health. All of the animals were handled according to approved institutional animal care and use committee (IACUC) protocols (#17-2560) of the University of Hawaii at Manoa. The protocol was approved by the Committee on the Ethics of Animal Experiments of the University of Hawaii at Manoa (Permit Number: A3423-01). All vital-dye imaging were performed under ice-cold MS222 anesthesia, and every effort was made to minimize suffering.

## Decision letter and Author response
Decision letter https://doi.org/10.7554/eLife.72463.sa1
Author response https://doi.org/10.7554/eLife.72463.sa2

# Additional files

## Supplementary files
• Supplementary file 1. Detailed statistical scores for *Figures 2, 3 and 6* and their figure supplements.
• Transparent reporting form
• Source data 1. Source data for the *Figures 1–6*.

## Data availability
All data generated and analyzed during this study are included in the supplementary Source Data file. Program scripts/codes are available in the public data depository (https://doi.org/10.5281/zenodo.5122894). All raw video data are available upon request. Sample video files are available at (https://doi.org/10.5281/zenodo.5122894).

The following dataset was generated:

| Author(s) | Year | Dataset title | Dataset URL | Database and Identifier |
|-----------|------|---------------|-------------|--------------------------|
| Motoko I | 2021 | Social-like responses are inducible in asocial Mexican cavefish despite the exhibition of strong repetitive behaviour | https://zenodo.org/record/5122894#.YPnDBR1ujsF | Zenodo, 10.5281/zenodo.5122894 |

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
