## [Decision Letter]

[Editors’ note: the authors submitted for reconsideration following the decision after peer review. What follows is the decision letter after the first round of review.]

Thank you for submitting your work entitled "Social-like responses are inducible in the asocial Mexican cavefish despite the exhibition of strong repetitive behavior" for consideration by *eLife*.

Your article has been reviewed by three peer reviewers, and the evaluation has been overseen by a Guest Editor and a Senior Editor. The following individual involved in the review of your submission has agreed to reveal their identity: Hans Hofmann (Reviewer #3).

Our decision has been reached after consultation between the reviewers and editors. Although the work is of interest, we regret to inform you that we have concluded that the findings at this stage are too preliminary for further consideration at *eLife*. That said, we would be willing re-consider a new submission at a later stage, if you were able to add the missing controls and address the other points raised. Please note that resubmission does not guarantee re-evaluation, let alone eventual acceptance.

Specifically, although you present a significant body of data, which supports many of your conclusions, there has been substantial concern about the lack of appropriate controls in order to rule out alternative explanations of the results. Furthermore, there have been questions about how some of the behaviours were defined and the interpretation of the relationship between social repetitive behaviours. Given that the behaviours described are influenced by speed and tendency to occupy central positions in fish, there have been concerns to what extend these behaviours would be representative of social behaviour, rather than kinematic differences in behaviour that are affected by a drug treatment already known to reduce movement speed.

Please note that we aim to publish articles with a single round of revision that under normal circumstances can be accomplished within two months. This means that articles that have potential, but in our judgment would need extensive additional work, will not be considered for revision. We do not intend any criticism of the quality of the data or the rigor of the science.

*Reviewer #1:*

Social-like responses are inducible in the asocial Mexican cavefish despite the exhibition of strong repetitive behavior by Iwashita and Yoshizawa examines social behavior in cavefish and their surface counterparts. While multiple previously published studies found a loss of schooling and shoaling behavior in cavefish compared to their highly social surface counterparts, as well as an essential role for vision in schooling in surface fish, through careful and thorough analysis, the authors overcome some of the limitations of previous studies to reveal that surface fish do indeed have social interactions in the dark, and that social interactions, while reduced, are indeed present in cavefish. Further, they find that in cavefish, the environment plays a role in these social interactions: these interactions increase in frequency and duration in a familiar environment. Further, they examine turning and find that both biased turning and social interactions are altered by a drug targeting the dopaminergic system in cavefish. Together, this study presents an interesting approach to examining social behavior in a comparative context, using novel methods in this context which allow these authors to uncover aspects of social behavior in a natural population of fish that were previously unknown.

The authors present a significant body of data, which supports many of their conclusions.

1. Additional information regarding how nearby interactions are defined would be helpful, along with an analysis to determine that these interactions are not simply occurring by chance, but rather signify social interactions.

2. Whether there is a relationship between turning bias (suggested here as a repetitive behavior) and social suppression is still unclear, and alternative explanations for the correlation between these behaviors in cavefish, should be discussed.

3. The authors do a great job of presenting many detailed analyses and explanations of how they performed these analyses. However, as structured, the amount of analysis presented in supplemental figures makes the manuscript difficult to follow in places. For example, there are six supplemental figures associated with Figure 3. However, many of these are unrelated to Figure 3, and there is a whole section of the paper from Lines 137-179 which only discusses these supplemental figures. Thus, some of these should be made into main figures for the paper. Similarly, there are 2 Supplementary files which describe methods that should go into main text.

4. The measurements of nearby interactions is crucial to the interpretations of the paper, and very interesting. However, I was a bit confused about how the authors chose their cutoffs for nearby interactions (for example, why was 4 seconds a better cut off than 3 seconds). Further, I was a little confused about whether some of these social interactions could occur by random chance. It seems like the authors could test this, as they have already performed "random" location permutation tests – are these close interactions overrepresented relative to what you would see if you analyzed these synthesized random videos? If this is the case, this would greatly strengthen their arguments.

5. The argument that there is a tradeoff between repetitive behavior and social behavior was weak – Couldn't these two traits just both be influenced by the pharmacological agent?

*Reviewer #2:*

The authors have presented a study that offers insight into a potentially fascinating question of social evolution. They examine two populations of *Astyanax* fish derived from surface or cave dwelling progenitors, comparing their behaviour in the absence of light shorter than 850nm (which they term 'dark', although this is not correct). Based on a previous finding that certain behaviours are reduced in the dark in this species, the authors sought to test whether other behaviours, which they term social-like interactions, are also diminished in the dark. They report 'numerous social-like' behaviours also occur in the dark in surface fish, and claim that treatment with a human anti-psychotic drug slowed velocity and restored some behaviours.

Although the aims are commendable, the experiment is poorly controlled and suffers from some major flaws. The first is in the usage and definitions of social behaviours. Most of the patterns of behaviours reported are associated with tendency to move towards or away from the centre of the tank. This is a crucial aspect as the 'social' behaviours reported by the authors are directly influenced by the tendency to mill in the centre of the tank, which has subsequent motion effects on turning frequency and inter-individual distance. Secondly, and most problematic for the interpretations of the authors is that although the authors claim there is no change in behaviour under their infra-red lighting conditions, the light sources they use emit at 850nm, which is within a known sensitivity range for many teleost species (Sensitivity Differences in Fish Offer Near-Infrared Vision as an Adaptable Evolutionary Trait Shcherbakov et al). Even other *Astyanax* studies use much longer wavelength light (e.g. 970nm). The most parsimonious explanation for almost all results is therefore that the surface fish can still see, and in fact the potentially strong side-welling light may have considerable consequences for their behaviour. The authors attempt to address this by presenting a visual stimulus (Video 4), but this stimulus is presented from above the water, whereas the light is side-welling and from below the water level. As such, it is not clear that this overhead stimulus would be visible in low-intensity, side-welling light in the assumed visible spectrum. Given the possibility that fish can see 850nm light, and the very poor controls conducted in this experiment to test for this, I simply do not accept the author's conclusion that the visual system plays no role in the behaviours described here. This problem is exacerbated in the main interpretations of the paper, which are the comparisons of swimming behaviour between cave and surface fish. A source of side-welling light would induce avoidance behaviour in sighted fish, and therefore also reduce the thigmotaxis commonly observed in this species. The finding that surface fish maintain more central positions in the tank is most likely a consequence of this avoidance, and without proper controls this remains the most parsimonious interpretation. That this effect was not observed in the cave fish, which certainly cannot see, provides more evidence for this interpretation. Because there is the potential that blind fish showed normal thigmotactic and exploratory behaviour of the entire tank, whereas surface fish were more centrally located in the tank (potentially because they avoided areas of strong side-welling light), then all subsequent analyses of inter-individual distance are questionable, since they are all impacted by the fact that surface fish are more central in the IR condition.

The relevant control would be to measure the tendency to avoid tank edges with natural light that is side-welling, but as can be seen in Video 3, the side-welling light is (surprisingly) removed with barriers, so no comparison can be made.

Unfortunately, I find this paper lacks the relevant controls that would be required to support the authors' interpretations.

*Reviewer #3:*

In the manuscript "Social-like responses are inducible in the asocial Mexican cavefish despite the exhibition of strong repetitive behavior," Iwashita and Yoshizawa examine to which extent the supposedly "asocial" blind cavefish, *Astyanax mexicanus*, exhibits "social-like interactions" depending on the familiarity of the environment. The authors show that such interactions exist, possibly regulated by the dopaminergic system.

Overall, this is a well-executed study that brings to bear advanced tracking technology to identify subtle behavioral phenotypes. In that sense it achieves its goals. To which extent those phenotypes have any adaptive value and whether the mechanisms underlying repetitive behavior might be shared between teleosts and mammals remains to be seen. In any case, the study should be of interest to readers in several fields.

(1) In Results, every experiment is introduced with a reference to similar human phenomena. I think it is fine to make this connection in the Introduction and Discussion, but it seems forced in the Results.

2) The authors may want to pay close attention to their use of terminology. It appears that words like "social," "collective," and "cooperative" are used interchangeably, where they clearly mean different things.

3) In the Discussion, the authors call on Friston's so-called "Free-Energy Principle" to interpret their results, which seems quite unnecessary. Also note that Friston's supposed principle rests on circular reasoning and cannot be experimentally falsified. I consider it unscientific.

4) The discussion of possible striatal mechanisms underlying the observed phenomena seems premature. The dopaminergic system was manipulated via systemic application of a drug (in aquarium water), so no inferences can be made about where the drug may have acted.

5) It is not clear that this study presents any evidence for the existence of "a deeply shared pathway that controls social-like and repetitive behaviors between teleost and mammals." The results presented here may well be consistent with this idea. However, to show this would require a phylogenetic comparative analysis, which was not the goal of the present study.

[Editors’ note: further revisions were suggested prior to acceptance, as described below.]

Thank you for submitting your article "Social-like responses are inducible in asocial Mexican cavefish despite the exhibition of strong repetitive behaviour" for consideration by *eLife*. Your article has been reviewed by two peer reviewers, and the evaluation has been overseen by a Guest Reviewing Editor and Christian Rutz as the Senior Editor. The reviewers have opted to remain anonymous.

While we are in principle happy to publish your article, please carefully address the remaining comments from Reviewer #2 in a final revision. Furthermore, please note that *eLife* has recently adopted the STRANGE framework, to help improve reporting standards and reproducibility in animal behaviour research. In your final revision, please consider scope for sampling biases and potential limitations to the generalisability of your findings, adding subject attribute data and discussion text as appropriate:

*Reviewer #1:*

The authors have done a good job with this revision. Their responses to the Reviewers' comments are thorough and convincing, and the revised manuscript reads well and is considerably strengthened. I think this will be a nice contribution.

*Reviewer #2:*

The authors addressed the majority of my concerns, and I was impressed by the extensive revisions and additional experiments and analysis they performed, and felt that these strengthened the manuscript.

1. The revised manuscript does an excellent job of explaining how nearby interactions were defined and why. However, this raised an additional question: The authors defined nearby interactions by identifying differences between simulated groups and actual groups of fish. The authors use IID (distance between each fish and every other fish in the group) to determine these cutoffs. However, the interactions are between only two fish, so it seems like NND would be the appropriate metric to use to determine what should be a nearby interaction, and this appears to be what the authors used in the last version of the manuscript. I recommend using pairwise interactions rather than group interactions to define what will ultimately be an analysis of interactions between pairs of fish.

2. The authors redoing experiments using IR light diffused across the tank resulted in many of the same conclusions as the previous manuscript. However, there were some differences in the reported results using this new method (notably, a lack of difference between cavefish and surface fish in NND and IID, differences in effects of the environment on position in the arena, etc.). Do the authors believe that the surface fish could see in the original set of assays but could not in the second, or do they have another explanation for these differences in results when the lights are moved to a different location?

---

## [Author Response]

[Editors’ note: the authors resubmitted a revised version of the paper for consideration. What follows is the authors’ response to the first round of review.]

[…] Specifically, although you present a significant body of data, which supports many of your conclusions, there has been substantial concern about the lack of appropriate controls in order to rule out alternative explanations of the results. Furthermore, there have been questions about how some of the behaviours were defined and the interpretation of the relationship between social repetitive behaviours. Given that the behaviours described are influenced by speed and tendency to occupy central positions in fish, there have been concerns to what extend these behaviours would be representative of social behaviour, rather than kinematic differences in behaviour that are affected by a drug treatment already known to reduce movement speed.

We took these concerns seriously and responded to all these points. Our responses are described in the cover letter. Briefly, we found that fish significantly slowed their swimming speed during social-like interactions (Figure 2G and H; Figure 3H and I; Figure 6E and F) and showed role switches between 'leader' and 'follower' during each interaction event (Figure 2—figure supplement 2). We further showed that these social-like events occurred in both peripheral and central areas of the recording arena (Figure 2—figure supplements 1 and 3) and showed no visual responsivity in the current setup. These results provide robust support that this detected behaviour represents an ‘affinity to each other’ and not unidirectional aggression or a result of avoiding peripheral areas.

Please note that we aim to publish articles with a single round of revision that under normal circumstances can be accomplished within two months. This means that articles that have potential, but in our judgment would need extensive additional work, will not be considered for revision. We do not intend any criticism of the quality of the data or the rigor of the science.Reviewer #1:[…] 1. Additional information regarding how nearby interactions are defined would be helpful, along with an analysis to determine that these interactions are not simply occurring by chance, but rather signify social interactions.

In this revision, we further described the detection method of the social-like interactions. Briefly, we filtered the nearby-interaction events by distance (below 5 cm) and duration (more than 4 s) based on the comparisons between actual and simulated data. The actual data were from 18 groups of social surface fish. The simulated data were generated by randomly sampling four individual fish’s X-Y coordinates (four fish per group in our experiment). Using these X-Y coordinates, we recalculated the inter-individual distances among each of the 1,280 simulated four-fish groups. The ranges of distance (≤ 3-6 cm) and duration (≥ 3-8 s) showed differences between the actual and simulated data. We chose a conservative set of cut-off values (i.e., ≤ 5 cm and ≥ 4 s). We also carefully classified the types of nearby interactions into the following categories: leading, dispersing, and approaching (Figure 2—figure supplement 2). As expected for reciprocal interactions, we observed role switches between leaders and followers within a single nearby-interaction bout event and between bouts within the same individual pair (Figure 2—figure supplement 2). This interaction can be categorized as fish in the “attraction” zone defined by Lukeman et al. (Lukeman et al., 2010). However, the fish could not form a collective cluster, probably due to the lack of long-distance sensing (visual sensing) in the dark. In fact, the surface fish could form a cluster under lighted conditions (Figure 2—figure supplement 3). We discussed these points in the Discussion section from the paragraph “Using our method of measuring nearby interactions via idTracker-based tracking…” (pg. 18) to the paragraph “In the dark, surface fish changed their swimming direction based on…” (pg. 20).

2. Whether there is a relationship between turning bias (suggested here as a repetitive behavior) and social suppression is still unclear, and alternative explanations for the correlation between these behaviors in cavefish, should be discussed.

We interpreted the reviewer’s point to mean that we cannot assume mutual exclusivity between turning bias and social-like interaction. We added a paragraph discussing the potential mechanisms of this antagonistic relationship (pg. 24). Briefly, turning bias or repetitive behaviour reduced the probability of social-like interactions, which could be due to a switch in the behaviour module in the striatal system or a simple fragmentation of social-like interaction events even though the individual may have the motivation to engage in social-like activities.

3. The authors do a great job of presenting many detailed analyses and explanations of how they performed these analyses. However, as structured, the amount of analysis presented in supplemental figures makes the manuscript difficult to follow in places. For example, there are six supplemental figures associated with Figure 3. However, many of these are unrelated to Figure 3, and there is a whole section of the paper from Lines 137-179 which only discusses these supplemental figures. Thus, some of these should be made into main figures for the paper. Similarly, there are 2 Supplementary files which describe methods that should go into main text.

Thank you very much for this constructive suggestion, and we fully agree with the reviewer’s point. We incorporated these supplemental data into the main figure (Figure 2) and simplified the logical flow.

4. The measurements of nearby interactions is crucial to the interpretations of the paper, and very interesting. However, I was a bit confused about how the authors chose their cutoffs for nearby interactions (for example, why was 4 seconds a better cut off than 3 seconds). Further, I was a little confused about whether some of these social interactions could occur by random chance. It seems like the authors could test this, as they have already performed "random" location permutation tests – are these close interactions overrepresented relative to what you would see if you analyzed these synthesized random videos? If this is the case, this would greatly strengthen their arguments.

Again, thank you very much for this significant advice. We introduced new paragraphs in the main body describing how we determined the cut-off parameters based on the comparisons between the actual and simulated randomly sampled data. The filtered nearby interactions (≤ 5 cm and ≥ 4 s) showed very similar behavioural modes—approaches, leading and following within the same bout—as in the lighted conditions with surface fish.

5. The argument that there is a tradeoff between repetitive behavior and social behavior was weak – Couldn't these two traits just both be influenced by the pharmacological agent?

We addressed this point above.

Reviewer #2:[…] Although the aims are commendable, the experiment is poorly controlled and suffers from some major flaws. The first is in the usage and definitions of social behaviours. Most of the patterns of behaviours reported are associated with tendency to move towards or away from the centre of the tank. This is a crucial aspect as the 'social' behaviours reported by the authors are directly influenced by the tendency to mill in the centre of the tank, which has subsequent motion effects on turning frequency and inter-individual distance. Secondly, and most problematic for the interpretations of the authors is that although the authors claim there is no change in behaviour under their infra-red lighting conditions, the light sources they use emit at 850nm, which is within a known sensitivity range for many teleost species (Sensitivity Differences in Fish Offer Near-Infrared Vision as an Adaptable Evolutionary Trait Shcherbakov et al). Even other Astyanax studies use much longer wavelength light (e.g. 970nm). The most parsimonious explanation for almost all results is therefore that the surface fish can still see, and in fact the potentially strong side-welling light may have considerable consequences for their behaviour. The authors attempt to address this by presenting a visual stimulus (Video 4), but this stimulus is presented from above the water, whereas the light is side-welling and from below the water level. As such, it is not clear that this overhead stimulus would be visible in low-intensity, side-welling light in the assumed visible spectrum. Given the possibility that fish can see 850nm light, and the very poor controls conducted in this experiment to test for this, I simply do not accept the author's conclusion that the visual system plays no role in the behaviours described here.

Thank you for pointing out these critical aspects. The first point, that is, that fish congregated in the centre to avoid ‘visible’ illumination from the side, regardless of their affinity for the others was rejected by looking at the swimming trajectories of the fish. The nearby interactions occurred in the centre and at the edges. Accordingly, we improved the visualization method for the fish trajectories (e.g., Figure 2—figure supplements 1 and 3).

*Astyanax* studies using 970-nm illumination are from one lateral line expert group, i.e., Dr. Sheryl Coombs laboratory. Additionally, thank you for sharing the reference regarding near-IR sensitivities in multiple teleost species. We re-designed the assay system and doublechecked that the sighted and non-sighted fish did not change their behaviour under our 850-nm illumination (Figure 1—figure supplement 1; Video 3). Considering a published report by Parry et al., where *Astyanax* retina microspectrophotometry showed no detectable absorption above 700 nm (Parry et al., 2003), we conclude that the current setup is dark based on *Astyanax* light sensitivity.

This problem is exacerbated in the main interpretations of the paper, which are the comparisons of swimming behaviour between cave and surface fish. A source of side-welling light would induce avoidance behaviour in sighted fish, and therefore also reduce the thigmotaxis commonly observed in this species. The finding that surface fish maintain more central positions in the tank is most likely a consequence of this avoidance, and without proper controls this remains the most parsimonious interpretation. That this effect was not observed in the cave fish, which certainly cannot see, provides more evidence for this interpretation. Because there is the potential that blind fish showed normal thigmotactic and exploratory behaviour of the entire tank, whereas surface fish were more centrally located in the tank (potentially because they avoided areas of strong side-welling light), then all subsequent analyses of inter-individual distance are questionable, since they are all impacted by the fact that surface fish are more central in the IR condition.The relevant control would be to measure the tendency to avoid tank edges with natural light that is side-welling, but as can be seen in Video 3, the side-welling light is (surprisingly) removed with barriers, so no comparison can be made.Unfortunately, I find this paper lacks the relevant controls that would be required to support the authors' interpretations.

Based on the comments by Reviewer 2, we systematically replicated all the experiments with a background LED light with a light diffuser. In this setting, surface fish and cavefish did not show a preference for the centre of the arena, did not show a preference for the illuminated or non-illuminated areas, and did not respond to the moving shade between the light source and fish tank. These facts supported the notion that surface fish did not respond to 850-nm light. In our original side-light setting, LEDs were naked without diffusers; thus, surface fish may have been able to sense the infrared light-LED itself or thermal energy. With the reported result of *Astyanax* retina’s microspectrophotometry (sensitive less than 700 nm; Parry et al., 2003), we conclude that our 850-nm infrared light did not affect the behaviours of either sighted surface fish or non-sighted cavefish.

Reviewer #3:[…] (1) In Results, every experiment is introduced with a reference to similar human phenomena. I think it is fine to make this connection in the Introduction and Discussion, but it seems forced in the Results.

Thank you for the suggestion. We removed most human-related statements from the Results section with the exception of the section on aripiprazole treatment. Aripiprazole treatment is justifiable only in relation to its use in humans as a medical treatment, due to its complex pharmacological profile and the history of its development.

2) The authors may want to pay close attention to their use of terminology. It appears that words like "social," "collective," and "cooperative" are used interchangeably, where they clearly mean different things.

This is a great suggestion. In the current revision, we avoided using “cooperative.” Additionally, we carefully chose the words “social-like” and “collective” based on the context.

3) In the Discussion, the authors call on Friston's so-called "Free-Energy Principle" to interpret their results, which seems quite unnecessary. Also note that Friston's supposed principle rests on circular reasoning and cannot be experimentally falsified. I consider it unscientific.

We removed reference to the “free-energy principle’; however, predictive coding is a well-accepted theory. We discussed a possible neural mechanism underlying biased turning based on predictive coding and a possible experimental approach to assessing the theory on pg. 22-23.

4) The discussion of possible striatal mechanisms underlying the observed phenomena seems premature. The dopaminergic system was manipulated via systemic application of a drug (in aquarium water), so no inferences can be made about where the drug may have acted.

We have expanded our discussion by taking into account the systemic effect of drug treatment. Now the sentence reads “Accordingly, bath treatment with a partial D2-dopamine receptor antagonist, aripiprazole, showed a parallel response between cavefish and humans, suggesting a potential shared pathway that controls social-like and repetitive behaviours in teleosts and mammals and possibly involves the striatum, POA, and/or other dopaminergic processing nuclei (Grillner, 2018; Kim et al., 2016; O’Connell and Hofmann, 2011)” (pg. 24).

5) It is not clear that this study presents any evidence for the existence of "a deeply shared pathway that controls social-like and repetitive behaviors between teleost and mammals." The results presented here may well be consistent with this idea. However, to show this would require a phylogenetic comparative analysis, which was not the goal of the present study.

We agree with the reviewer and eliminated phylogeny-related statements. This statement now reads “the presented results suggest that the asocial cave population is still capable of performing social-like nearby interactions under environmental or pharmacological treatments and shares an antagonistic association between social-like and repetitive behaviours with mammals.” (pg. 6).

[Editors’ note: what follows is the authors’ response to the second round of review.]

Reviewer #2:The authors addressed the majority of my concerns, and I was impressed by the extensive revisions and additional experiments and analysis they performed, and felt that these strengthened the manuscript.1. The revised manuscript does an excellent job of explaining how nearby interactions were defined and why. However, this raised an additional question: The authors defined nearby interactions by identifying differences between simulated groups and actual groups of fish. The authors use IID (distance between each fish and every other fish in the group) to determine these cutoffs. However, the interactions are between only two fish, so it seems like NND would be the appropriate metric to use to determine what should be a nearby interaction, and this appears to be what the authors used in the last version of the manuscript. I recommend using pairwise interactions rather than group interactions to define what will ultimately be an analysis of interactions between pairs of fish.

Thank you for addressing this critical point with thoughtful comments. We strategically chose IID (average distance between each pair of fish) instead of NND (nearest neighbour distance) because we intended to measure nearby interactions (one-by-one) while two, three, or four fish were interacting with each other. This allowed us to count the social-like events that we carefully investigated when the filtered social-like events matched what we observed in fish behaviours in each recorded video. For example, suppose fish A and fish B are interacting with each other, and fish C passes by fish A more closely than it passes by fish B. With the cut-off based on IID, the fish A-fish B interaction is continuously counted while fish C passes by fish A because this cut-off is based on the average distance of each pair of fish (Figure 2—figure supplement 1A). The fish A-fish C interaction may not be counted because of a shorter duration than the cut-off period. In contrast, by using the NND cut-off, three events are counted separately: (1) the fish A-fish B interaction before fish C passes by, (2) the fish A-fish C interaction, and (3) the fish A-fish B interaction after fish C passes by (note, NND is shorter than IID). This may exclude the fish A-fish B interaction since each of durations (1) and (3), not the sum of (1) and (3), can be shorter than the duration cut-off.

Again, we are reporting nearby interaction durations/bouts of fish pairs within a cluster of fish (i.e., shorter distances than the IID-based cut off). The NND-based cut-off fragments the nearby interaction bout and reduces the S/N ratio (the passing-by events are noise).

To clarify this point, we added the sentence "IID instead of NND was strategically chosen because we were interested in interactions not only between two fish (NND) but also among three or four fish (IID) that were closer than expected by chance” to the Results section on pg 8.

2. The authors redoing experiments using IR light diffused across the tank resulted in many of the same conclusions as the previous manuscript. However, there were some differences in the reported results using this new method (notably, a lack of difference between cavefish and surface fish in NND and IID, differences in effects of the environment on position in the arena, etc.). Do the authors believe that the surface fish could see in the original set of assays but could not in the second, or do they have another explanation for these differences in results when the lights are moved to a different location?

Whether the fish could "see or not see" is an important point. We made this point only in response to Reviewer 2 in our resubmission cover letter because it is not necessary for the reader to know about our original side-light setting. What we wrote in our response was, "In our original side-light setting, LEDs were naked without diffusers; thus, surface fish may have been able to sense the infrared light-LED itself or thermal energy."

We hope this answers your question.